



# Uncertainty Characterization of HOAPS-3.3 Latent Heat Flux Related Parameters

Julian Kinzel[1], Marc Schröder[1], Karsten Fennig[1], Axel Andersson[2], and Rainer Hollmann[1]

[1]Satellite-Based Climate Monitoring, Deutscher Wetterdienst, Frankfurter Strasse 135, 63067 Offenbach, Germany
[2]Marine Data Center, Deutscher Wetterdienst, Bernhard-Nocht-Straße 76, 20359 Hamburg, Germany

*Correspondence to:* Julian Kinzel, Email: julian.kinzel@dwd.de

**Abstract.** Latent heat fluxes ($LHF$) are one of the main contributors to the global energy budget. As the density of LHF measurements over the global oceans is generally poor, the potential of remotely sensed LHF for meteorological applications is enormous. However, to date none of the available satellite products include estimates of systematic, random retrieval, and sampling uncertainties, all of which are essential for assessing their quality. Here, this challenge is taken on by applying region-
5    ally independent multi-dimensional bias analyses to $LHF$-related parameters (wind speed $U$, near-surface specific humidity $q_a$, and sea surface saturation specific humidity $q_s$) of the Hamburg Ocean Atmosphere Parameters and Fluxes from Satellite (HOAPS) climatology. In connection with multiple triple collocation analyses, it is demonstrated how both instantaneous (gridded) uncertainty measures may be assigned to each pixel (grid box). A high-quality *in situ* data archive including buoys and selected ships serves as the ground reference. Results show that systematic $LHF$ uncertainties range between 15-50 W
10    m$^{-2}$ with a global mean of 25 W m$^{-2}$. Local maxima are mainly found over the subtropical ocean basins as well as along the western boundary currents. Investigations indicate that contributions by $q_a$ ($U$) to the overall $LHF$ uncertainty are in the order of 60 % (25 %). From an instantaneous point of view, random retrieval uncertainties are specifically large over the subtropics with a global average of 37 W m$^{-2}$. In a climatological sense, their magnitudes become negligible, as do respective sampling uncertainties. Time series analyses show footprints of climate events, such as the strong El Niño during 1997/98. Regional
15    and seasonal analyses suggest that largest total (i.e., systematic + instantaneous random) $LHF$ uncertainties are seen over the Gulf Stream and the Indian monsoon region during boreal winter. In light of the uncertainty measures, the observed continuous global mean $LHF$ increase up to 2009 needs to be treated with caution. First intercomparisons to other $LHF$ climatologies (*in situ*, satellite) reveal overall resemblance with few, yet distinct exceptions.





## 1 Introduction

Exchanges of energy and moisture at the atmosphere–ocean interface represent a critical coupling mechanism within the climate system. Roughly 20 % of the total radiation absorbed by the Earth's surface is transferred back to the atmosphere by means of turbulent heat fluxes (Trenberth et al., 2009). Specifically, latent heat fluxes ($LHF$) significantly control the
surface energy budget and are, next to radiative fluxes, one of the main contributors to heating and cooling of the oceans. Approximately 86 % of the global evaporation occurs over the ocean basin (Baumgartner and Reichel, 1975), demonstrating that this water and oceanic energy transfer is a key component of the overall Earth's energy budget. The fifth assessment report of the Intergovernmental Panel on Climate Change (IPCC) emphasizes the role of heat transfer between ocean and atmosphere in driving the oceanic circulation. It stresses that flux anomalies can impact water mass formation rates and alter oceanic and
atmospheric circulation (IPCC, 2013) due to its influence on sea surface salinity and thus the ocean surface's density (e.g. Grodsky et al., 2009). Additionally, $LHF$ modifies the atmospheric stability distribution and triggers convection, which in turn strongly impacts cloud formation and precipitation. Next to its impact on oceanic processes, this highlights the important role of $LHF$ in modulating the atmospheric circulation on a variety of scales.

To improve our understanding of global energy and water cycle variability as well as model simulations of climate variations,
it is of great importance to accurately measure $LHF$ over the global oceans at the highest possible resolution (e.g. Chou et al., 2004). The need for accurate surface fluxes has, for example, been picked up by the World Climate Research Programme (WCRP), the WCRP Global Energy and Water Cycle Experiment (GEWEX), and the Climate Variations (CLIVAR) Science Steering Group (e.g. Curry et al., 2004). This is ideally achieved through accurate observations and correct implementations of parameterizations in coupled models. Liu and Curry (2006), for example, stress that accurate $LHF$ are essential for a correct
forcing of ocean models and for evaluating numerical weather prediction. Additionally, reliable long-term global $LHF$ data sets represent a substantial input to assimilation experiments, for instance the oceanic synthesis performed by the German contribution to Estimating the Circulation and Climate of the Ocean (GECCO, GECCO2, e.g. Köhl and Stammer, 2008; Köhl, 2015). Such syntheses allow for capturing variability and trends in the turbulent exchange processes, which may exert changes to the entire climate system.

Several $LHF$ data records exist, which differ in satellite instrumentation, creation process, data density, as well as spatial and temporal extent. These are either based on *in situ* measurements, reanalysis or remotely sensed data. Apart from isolated direct *in situ* measurements using e.g. sonic anemometers, all data sources have in common that bulk flux algorithms are applied to derive $LHF$. The near-surface wind speed ($U$), the saturation specific humidity at the sea surface ($q_s$), and the near-surface specific humidity ($q_a$) serve as input bulk parameters, on which the parameterized $LHF$ primarily depend.

However, global $LHF$ time series are often subject to uncertainties of unknown magnitudes, which for example hampers the conclusion whether there is a significant multi-decadal trend in global $LHF$ or not (e.g. Chapter 3.4.2, IPCC, 2013). On the one hand, *in situ* $LHF$ climatologies, which often include data from buoys and ships, are known to contain biases (e.g. Wang and McPhaden, 2001), to be of variable quality, and to be unevenly sampled. Although research vessel measurements of e.g. $q_a$ are expected to be of good quality (e.g. Roberts et al., 2010), they are regionally limited, which also accounts





for data from moored buoys (Weller et al., 2008). Issues related to poor data densities over the Southern Ocean, amongst others, are for example stressed in Yu and Weller (2007), Bourassa et al. (2013), and Prytherch et al. (2014). In consequence, this impedes a meaningful discussion regarding the quality of $LHF$ in this climatologically important region (Josey, 2011). Despite the above-addressed issues, the research community has put effort into uncertainty characterizations regarding *in situ*

$LHF$-related measurements. Whereas random uncertainties of ship-based $LHF$-related parameters are for example discussed in Gleckler and Weare (1997), Kent and Berry (2005), and Kent and Taylor (2006), systematic uncertainties are assessed in e.g. Kent et al. (1993) and Kent and Taylor (1996). An example of an *in situ* $LHF$ climatology incorporating uncertainty estimates is given by NOCS v2.0 (Berry and Kent, 2009).

On the other hand, global reanalysis products such as ERA-Interim (Dee et al., 2011) and NCEP-NCAR (Saha et al., 2010)

have a high temporal resolution and extent of time series, but are not capable of resolving local-scale processes due to a lack of spatial detail (Winterfeldt et al., 2010). Specifically over data-sparse regions, more weight is given to the atmospheric model, which is also prone to uncertainties (e.g. Gulev et al., 2007). At some level, atmospheric reanalysis thus suffer from problems in their freshwater budgets (e.g. Schlosser and Houser, 2006; Trenberth et al., 2007).

Several remote sensing data records incorporate $LHF$-related parameters, e.g. the Japanese Ocean Flux data sets with Use

of Remote Sensing Observations (J-OFURO) satellite climatology (Kubota et al., 2002), the Goddard Satellite-based Surface Turbulent Heat Flux (GSSTF) Version 3 product (Shie et al., 2012), the updated version of the French Research Institute for Exploitation of the Sea (IFREMER) turbulent flux estimates (Bentamy et al., 2013), the SeaFlux Version 1 and 2 data sets (Clayson et al., 2015), and the Hamburg Ocean Atmosphere Parameters and Fluxes from Satellite (HOAPS) climatology (Andersson et al., 2010; Fennig et al., 2012). As all incorporate data with high spatial resolution and cover up to several

decades, they have a vast potential for climate research applications.

The HOAPS data set is a completely satellite-based, single-source climatology of precipitation, evaporation, related turbulent heat fluxes, and atmospheric state variables over the global ice-free oceans. The usefulness of the HOAPS climatology for climatological applications has been tested among numerous intercomparison studies and promising results have been published within Bentamy et al. (2003), Bourras (2006), Klepp et al. (2008), Winterfeldt et al. (2010), Andersson et al. (2011),

and Stendardo et al. (2016). In the framework of assessing sea surface freshwater fluxes, Romanova et al. (2010) for example conclude that HOAPS-3 is well suited for global applications and serves as an important and independent data set that should be included in future ocean synthesis.

As *in situ* and reanalysis data records, remotely sensed $LHF$ climatologies are also prone to uncertainties. Next to calibration uncertainties and aliasing problems (Bentamy et al., 2003), uncertainty sources either originate from uncertainties in the

parameterization (Zeng et al., 1998; Brunke et al., 2002, 2003) or may be linked to the inaccuracy of the input bulk variables (Bourassa et al., 2013). In the framework of an oceanic $LHF$ assessment, Brunke et al. (2011) for example conclude that the uncertainty of HOAPS-3 $LHF$ is largely composed of bulk variable-caused issues due to inaccuracies of their individual retrievals. Liu and Curry (2006) reason similarly, while assessing discrepancies of remotely sensed and reanalysis $LHF$ during the 1990s. Romanova et al. (2010) recall that specifically early satellite-based products contain large uncertainties, as also

shown by investigations regarding the hydrological cycle by Mehta et al. (2005). The knowledge of both accuracy and preci-





sion of the remotely sensed bulk parameters is critical for assessing the quality of satellite-based $LHF$, as the uncertainties propagate through the applied $LHF$ bulk formula. Finally, irregular sampling from space introduces sampling uncertainties, which may locally become substantial (e.g. Gulev et al., 2007).

To better quantify the quality of satellite-based data sets, Prytherch et al. (2014) recently emphasized the value grid box
based, comprehensive uncertainty estimates (in their case of $q_a$) would have for climate research purposes. To date, none of the above-listed, satellite-based data sets are accompanied by $LHF$-related uncertainty estimates, which hampers a quality assessment of the air–sea fluxes and related parameters. Such uncertainty assessments would go beyond conventional $LHF$ intercomparison studies (as e.g. presented in Chou et al., 2004; Yu et al., 2011), as they would allow for quantifying the data's accuracy and precision. Uncertainty evaluations in ocean surface turbulent fluxes has for example been carried out by Brunke
et al. (2011), who decomposed overall biases with respect to direct *in situ* records into a bulk variable and a residual component, the latter which also includes the measurement uncertainty. A current overview study by Loew et al. (2017) highlights the necessity of earth observation data validation and pools different approaches across communities. Finally, assimilation schemes like GECCO require such uncertainty information prior to assimilating respective fields in ocean models.

In the framework of the German research initiatives 'FOR1740' and 'FOR21740' ('Atlantic Freshwater Cycle', http://
for1740.zmaw.de/), the lack of uncertainty information is overcome by taking on the challenge of quantifying systematic, random, and sampling uncertainties inherent to HOAPS-3.3 $LHF$-related parameters. Rigorous error propagation to the instantaneous $LHF$-related data is performed, which accounts for how uncertainties in the bulk parameters propagate into uncertainties of $LHF$ themselves, while accounting for covariances of the contributing parameters.

Section 2 presents the applied data sources in more detail. As to systematic uncertainty patterns, the approach of double
collocation and multi-dimensional bias analyses is introduced in Sect. 3. This is complemented by the strategy of decomposing random uncertainties via multiple triple collocation to separate the eligible random satellite retrieval uncertainty from collocation and *in situ* measurement contributions. All uncertainty components are presented in Sect. 4, where seasonal and regional differentiations allow for assessing the uncertainty spectrum. This is supplemented by trend analysis in light of the derived uncertainty estimates. Section 5 provides a summary and a brief outlook regarding ongoing work.

## 2   Data

### 2.1   HOAPS-3.3 Pixel-Level Data Records

Apart from the sea surface temperature ($SST$), all HOAPS parameters are derived from intercalibrated Special Sensor Microwave/Imager (SSM/I) and Special Sensor Microwave Imager/Sounder (SSMIS) passive microwave radiometers, which are
installed aboard the polar orbiting satellites of the United States Air Force Defense Meteorological Satellite Program (DMSP). HOAPS provides consistently derived global fields of freshwater flux related parameters. Regarding sensor specifications and orbital paths, the reader is referred to e.g. Andersson et al. (2010).



Here, the focus lies on the HOAPS-3.3 pixel-level, which has been produced as an extension to the HOAPS-3.2 data set (Andersson et al., 2010; Fennig et al., 2012) in the framework of the ongoing DFG research activity. It covers the time period from 1987 to 2015, during which a total number of nine satellite instruments were in operational mode. Compared to HOAPS-3.2, HOAPS-3.3 has been temporally extended up to 2015 and is based on a pre-release of the CM SAF SSM/I and SSMIS FCDR.

This reprocessing included a homogenization of the radiance time series by means of an improved inter-sensor calibration with respect to the DMSP F11 instrument. Earth incidence angle normalization corrections were applied, following a method described by Fuhrhop and Simmer (1996). Its extensive documentation is available online (Fennig et al., 2013). Since the HOAPS-3.1 release, HOAPS is hosted by the EUMETSAT Satellite Application Facility on Climate Monitoring (CM SAF), whereupon its further development is shared with the University of Hamburg and the Max Planck Institute for Meteorology

(Hamburg). In this study, the pixel-level HOAPS-3.3 data in sensor resolution is used, which implies that no aggregation for gridding purposes has been applied.

HOAPS-3.3 $q_a$ relies on a direct, four-channel retrieval algorithm by Bentamy et al. (2003), which is based on a modified version of the two-step multi-channel regression model by Schulz et al. (1993) and its refinement by Schlüssel (1996). 1000 globally collocated pairs of SSM/I TBs and ship data between 1996-1997 were used to estimate the new values for the

coefficients in the Schulz model.

To account for the non-linearity of the problem, the HOAPS-3.3 $U$ algorithm uses a neural network approach with three layers after Krasnopolsky et al. (1995) to derive the wind speed at 10 m above sea level (a.s.l.). The network was trained with a composite data set of buoy measurements, which was compiled using matchups of SSM/I F11 brightness temperatures (TBs) and near-surface wind speed measurements from the National Oceanographic and Atmospheric Administration (NOAA)

National Data Buoy Center (NDBC) and the Tropical Atmosphere Ocean (TAO) array between 1997-98. Radiative transfer simulations based on radiosonde profiles served as input for the training data set (Andersson et al., 2010).

HOAPS-3.3 $SST$ is based on the AVHRR Pathfinder Version 5.2 and is obtained from the US National Oceanographic Data Center and the Group for High Resolution Sea Surface Temperature (http://pathfinder.nodc.noaa.gov). The data are an updated version of the Pathfinder Version 5.0 and 5.1 collection described in Casey et al. (2010). A static bias correction of +0.17 K has

been applied to HOAPS-3.3 $SST$ data in order to revert the Pathfinder Version 5.2 skin correction and thus achieve consistency with Version 5.0 used in HOAPS-3.2.

HOAPS-3.3 sea surface saturation specific humidity $q_s$ is derived by applying the Magnus formula (Murray, 1967) to $SST$, while accounting for a constant salinity correction factor of 0.98. Zeng et al. (1998), e.g., showed that omitting the factor under strong wind conditions has a significant impact on resulting $LHF$.

HOAPS-3.3 $LHF$ is based on the Coupled Ocean–Atmosphere Response Experiment (COARE) 2.6a bulk flux algorithm. With minor modifications of physics and parameterizations, the algorithm is published as COARE 3.0a by Fairall et al. (2003). $U$, $q_s$, and $q_a$ are required as input. The latter depends on the surface air temperature, which is estimated by assuming a constant relative humidity of 80 % (Liu et al., 1994) and air-sea temperature difference of 1 K (Wells and King-Hele, 1990). COARE-3.0 is widely accepted within the scientific community; its benefits are for example presented in the framework of an

intercomparison study by Brunke et al. (2003).



## 2.2 DWD-ICOADS Data Archive

Hourly *in situ* measurements of $U$, $q_s$, and $q_a$ (bulk parameters, as of now) have been provided by the Marine Climate Data Center of the German Meteorological Service (DWD), supervised by the Marine Meteorological Office (Seewetteramt, SWA). While data prior to 1995 is excluded due to a comparatively poor *in situ* data coverage, the data set used here includes measure-

ments up to 2008. It comprises global high-quality shipborne measurements as well as data provided by drifting and moored buoys. In case of data gaps within the SWA archive, the *in situ* data basis was extended at SWA by available International Comprehensive Ocean–Atmosphere Data Set (ICOADS) measurements (Version 2.5, Woodruff et al., 2011). These records contain hourly global measurements obtained from ships, moored and drifting buoys as well as near-surface measurements of oceanographic profiles. Several quality checks were performed at SWA prior to using the merged DWD-ICOADS data, which

resulted in quality index assignments to each observation. Details regarding the flagging procedures carried out at SWA are given in Kinzel et al. (2016).

In preparation for the uncertainty analyses, further filtering and correcting procedures to both ship and buoy data were carried out. Regarding ship records, annual lists of Voluntary Observing Ships (VOS) metadata (Kent et al., 2007) were employed. Most of the supplementary buoy metadata was extracted from the Data Buoy Cooperation Panel, which particularly includes a

fleet of moored buoy arrays operated by NDBC. Metadata of the Global Tropical Moored Buoy Array, such as TAO-TRITON (Pacific-), PIRATA (Atlantic-), and RAMA (Indian Ocean) were obtained from the Pacific Marine Environment Laboratory (PMEL).

ICOADS VOS estimates of $q_a$ are based on wet bulb temperature measurements, typically using mercury thermometers, which are often exposed in either (ventilated) screens or sling psychrometers (Kent et al., 2007). $q_a$ is eventually derived by

applying the psychrometric formula. By contrast, $q_a$ estimates of buoys originate from measurements of air temperature and relative humidity. For this study, $q_a$ of both VOS and buoys were not corrected to the HOAPS-3.3 reference of 10 m a.s.l., assuming neutral stratification. A discussion related to this approach is published in Kinzel et al. (2016). It is in line with Prytherch et al. (2014), who conclude that a conversion to 10 a.s.l. (neutral stability) substantially adds to the noise in the resulting *in situ* $q_a$. The aspect of correcting $q_a$ with respect to height and stratification is also elucidated in Bentamy et al.

(2003) and correction effects are presented in Kent et al. (2014).

DWD-ICOADS VOS $U$ are either measured using anemometers (likewise for buoys) or are estimated from the sea state, depending on the preference of the country recruiting the VOS (Kent et al., 2007). By means of the measured wind speed and direction, the true wind speeds are derived considering the ship's speed and direction. If a specific anemometer height was not given, it was estimated from the annual global mean height difference with respect to the thermometer platform. For each year,

this single height difference value is based on all contributing ship records with complete metadata information. Prior to 2002, no thermometer heights were available; consequently, the height difference was set to 6 m (average between 2002-2008). In case both sensor heights were unknown, the linear fits shown in Table 4 of Kent et al. (2007) were used to derive anemometer heights based on available ship length metadata. It was assumed that these ship type dependent linear fits (Kent et al., 2007, their Fig. 11) introduce negligible uncertainties to the sensor height derivation. Given the anemometer heights of both VOS and



buoys, *in situ* wind speeds were corrected to the HOAPS-3.3 standard height of 10 m a.s.l. to remove inhomogeneities, using the iterative equivalent neutral stability approach of Fairall et al. (2003). With the exception of e.g. (stable stratified) upwelling regimes or local instabilities, the equivalent neutral stability assumption is valid over vast regions of the open oceans. The correction using a neutral wind equivalent profile has been suggested by e.g. Shearman and Zelenko (1989). It is argued that in

case of VOS, the omission of a correction would lead to a positive wind speed bias, as the average wind sensor height is given by 18 m (Kent et al., 2014). By contrast, buoy $U$ would be low-biased.

    VOS $SST$ measurement techniques differ in terms of platform, measurement depth, and extent of automation. Strictly speaking, *in situ* $SST$ are sub-surface temperatures and thus differ from the HOAPS-3.3 Pathfinder $SST$, which are treated as a skin $SST$ for the surface flux calculations. This necessitates an *in situ* cool-skin correction as a function of wind speed, following

Donlon et al. (2002). Their Equation (2) was applied, omitting all records subject to wind speeds below 2 m s$^{-1}$ (corrected to 10 m a.s.l.), as the exponential fit introduces additional uncertainty for very calm conditions. On average, the $SST$ correction reduced the DWD-ICOADS $SST$ by approximately 0.17 K. Moreover, the warm layer part of the COARE 3.0 algorithm is not implemented in HOAPS-3.3 due to the lack of a continuous diurnal cycle information on the surface radiation budget from the SSM/I and SSMIS measurements. To be directly comparable to the *in situ* counterpart, all *in situ* measurements taken during

local daytime were excluded. As only night-time *in situ* measurements during non-calm conditions were considered, the sea water temperature gradient within the uppermost meters of the water column is thought to be negligible. A $SST$ correction with respect to the sensor depths was therefore omitted for both VOS and buoys, independent of the measurement platform.

    All VOS data processing described above were carried out for research vessels (so-called 'special ships') and merchant vessels only due to vast data amounts and in order to minimize *in situ* uncertainties. In case of multiple triple collocation

analysis (Sect. 3.4), buoy records were excluded to ensure having a consistent, globally distributed data set as the ground reference for the random decomposition procedure. It is argued that the vast amount of remaining triplets authorizes this restriction.

    Despite strict filtering and correcting procedures, *in situ* measurement uncertainties related to sensor types, measurement heights and positions, and solar radiation contamination may remain (e.g. Bourassa et al., 2013). Assessments regarding the

quality of the reference data are beyond the scope of this article. The *in situ* data basis is therefore considered as the bias-free, ground reference.

## 3   Methodology

This Section describes the technical background for deriving systematic, random, and sampling uncertainties inherent to HOAPS-3.3. The uncertainties will be examined from an either instantaneous or climatological point of view, depending

on the scale of interest and thus the application. The random uncertainty decomposition presented in Kinzel et al. (2016) is therefore complemented, leading to a complete HOAPS-3.3 uncertainty characterization of $LHF$-related parameters.





### 3.1 Double Collocation Analysis

In preparation for uncertainty calculations, a double collocation analysis is performed for the time period of 2001–2008, resulting in paired matchups of $LHF$-related HOAPS-3.3 and *in situ* data. Although HOAPS-3.3 lasts until 2015, collocations between 2009–2015 were not performed, as the DWD-ICOADS data archive only lasts until 2008. The collocated pairs are
based on the so-called nearest neighbor approach; that is, HOAPS-3.3 pixels are assigned to respective *in situ* observations closest in time and space. Parameter-independent collocation criteria of $\Delta x = 50$ km and $\Delta t = 60$ min are chosen. These are more restrictive than those derived in e.g. Kinzel (2013). Due to the vast amount of available matchups this is justifiable and ensures that e.g. strong spatial and/or temporal gradients associated with fronts are discarded from further analysis.

Figure 1a-d exemplarily shows scatter density plots of the $q_a$ bias (2001-2008) as a function of the atmospheric state param-
eters $q_a$ ("hair"), $U$ ("wind"), $SST$ ("asst"), and water vapour path ("wvpa"), resulting from the double collocation analyses. Overall, 13.8 million matchups contribute to each subplot. Figure 1a indicates that HOAPS-3.3 overestimates near-surface specific humidities for $q_a$ between 7–12 g kg$^{-1}$ and in the inner tropics ($q_a \approx 20$ g kg$^{-1}$). In return, biases are negative over Arctic (< 5 g kg$^{-1}$) and subtropical (12–17 g kg$^{-1}$) humidity regimes. The latter regions is also subject to largest random uncertainties, which exceed 2 g kg$^{-1}$. See Kinzel et al. (2016) for more details on the analysis of HOAPS-3.3 $q_a$. The humidity bias
and standard deviation dependency on $SST$ (Fig. 1c) shows similar features regarding regimes of over- and underestimation. Humidity biases as a function of wind speed are illustrated in Fig. 1b. The distribution is somewhat linear, where low (high) wind regimes are over- (under-) represented in HOAPS-3.3. In contrast to the remaining atmospheric state parameters, the random uncertainty decreases fairly linearly with increasing wind speeds. A dependency of $dq_a$ on the total integrated water vapour (Fig. 1d) shows only few distinct features. Most matchups coincide with values below 20 kg m$^{-2}$. With the exception of
smallest values, these result in positive biases with respect to HOAPS-3.3.

A comparison of e.g. Fig. 1a and b indicates that the simple one-dimensional bias analyses may be misleading when it comes to HOAPS-3.3 $q_a$-related uncertainty characterizations. Average $q_a$ off the Arabian Peninsula, for example, are in the order of 14–15 g kg$^{-1}$. According to Fig. 1a, this is associated with a HOAPS-3.3 $q_a$ underestimation. At the same time, climatological mean wind speeds are as low as 3–5 m s$^{-1}$, which goes along with a HOAPS-3.3 $q_a$ overestimation (Fig.
1b). This is no contradiction, but rather indicates that the HOAPS-3.3 $q_a$ retrieval seems to encounter challenges for specific humidity and wind regimes. Furthermore, a constraint to one-dimensional analyses implies for example that parts of the random uncertainties illustrated in Fig. 1a (bars) receive a systematic component in Fig. 1b (squares). This conclusion motivates to proceed with multi-dimensional bias analyses, where all possible atmospheric states, i.e. combinations of the four chosen atmospheric state parameters, are accounted for simultaneously. This approach finally allows for separating systematic from
random uncertainties. Results illustrated in Fig. 1 can therefore be considered as a preliminary stage of the four-dimensional bias analyses introduced in Sect. 3.2, where each of the four atmospheric state variables (i.e., Fig. 1, x-axes) represent one dimension. Analogously to Fig. 1, one-dimensional analyses are performed for both $dU$ and $dq_s$ (not shown).



## 3.2 Multi-Dimensional Bias Analyses

The bulk formula for $LHF$ is given by

$$LHF = \rho_a L_V C_E U (q_s - q_a), \tag{1}$$

where $\rho_a$ is the density of moist air and $L_V$ the latent heat of vaporization. $\rho_a$ is derived as a function of HOAPS-3.3 $q_a$ and
near-surface air temperature. Likewise, $L_V$ is computed simultaneously as a function of HOAPS-3.3 $SST$.

Assuming uncertainties in $\rho_a$ and $L_V$ to be negligible and according to standard error propagation, the overall $LHF$ uncertainty is a function of the systematic and random uncertainties introduced by the remaining parameters. In case of $U$, $q_s$, and $q_a$ these are assumed to depend on the concurrent atmospheric state. The combination of $q_a$, $U$, $SST$, and water vapour path is thought to represent the concurrent atmospheric state best. Therefore, the one-dimensional consideration presented in
Sect. 3.1 is expanded by creating four-dimensional look up tables (LUTs) including $20^4$ entries, respectively. The dimension is reflected in the exponent, whereas its base represents the amount of bins per dimension. The values of all four dimensional vectors are essential for assigning instantaneous, absolute differences (HOAPS-3.3 minus *in situ*) to the correct LUT bin and are predetermined by the respective x-values of the black squares shown in Fig. 1a-d.

The uncertainty dependency on specific ambient conditions overcomes the issues introduced by data-sparse regions, such as
the Southern Ocean and the tropical oceans (e.g. Kent and Berry, 2005). Here, it is knowingly turned away from the dependency on matchup density, which implies that the LUTs are valid on a global scale. Due to the immense data availability, their input biases are confined to matchups from 2001–2008 ($dq_a$, $dU$) and 1998–2001/2006–2008 ($dq_s$). A thorough elucidation of the multi-dimensional bias analysis is presented in Kinzel et al. (2016), exemplarily for HOAPS-3.2 $q_a$ (Sect. 2c and Fig. 5, left therein). Here, it is applied to all three bulk parameters, which results in both systematic and *total* random uncertainty LUTs.
As to the Dalton Number $C_E$, the estimates of Fairall et al. (2003) are applied by assigning 5 % (10 %) of systematic uncertainty of $C_E$ for wind speeds smaller (larger) than 10 m s$^{-1}$. For even stronger wind speeds, the estimate of Gleckler and Weare (1997) of 12 % is taken on. Independently of $U$, random uncertainties of 20 % are assigned, as proposed by Gleckler and Weare (1997).

Recall that the aim is to characterize *uncertainty* and not bias patterns. This implies that *absolute* systematic uncertainty
values are generally presented, i.e., magnitudes are invariably positive. Results presented in Sect. 4.4-4.6 can therefore be considered as illustrating the upper boundaries of systematic (that is, climatological) uncertainties.

## 3.3 HOAPS-3.3 Uncertainty of LHF

The uncertainties in $LHF$ are caused by uncertainties in all bulk input parameters contributing to Eq. (1). Assuming the underlying parameterizations to be correct, $LHF$ uncertainties can thus be derived by carrying out standard error propagation.
These uncertainty estimates are assigned at each point in time and space.



Total instantaneous $LHF$ uncertainties, $\sigma_{LHF}$, are derived as follows:

$$\sigma_{LHF} = \sqrt{\left(\frac{\partial LHF}{\partial x}\right)^2 \sigma_x^2 + \left(\frac{\partial LHF}{\partial y}\right)^2 \sigma_y^2 + 2r_{xy}\left(\frac{\partial LHF}{\partial x}\frac{\partial LHF}{\partial y}\right)\sigma_x\sigma_y}, \tag{2}$$

where $x$ and $y$ are place holders of $U$, $q_s$, $q_a$, and $C_E$. $r_{xy}$ is the correlation coefficient between $x$ and $y$. For each combination of $x$ and $y$, the average of daily global mean correlation coefficients between 1995 and 2008 is applied.

$\sigma_x$ and $\sigma_y$ are *total* uncertainties in $x$ and $y$. These can be decomposed into systematic and random components:

$$\left(\frac{\partial LHF}{\partial x}\right)^2 \sigma_x^2 \cong \left(\frac{\partial LHF}{\partial x}\right)^2 \sigma_{x,sys}^2 + \left(\frac{\partial LHF}{\partial x}\right)^2 \sigma_{x,ran}^2 \left(N^{-1/2}\right)^2. \tag{3}$$

$N$ is the number of HOAPS-3.3 satellite observations (N=1 for instantaneous $LHF$ uncertainties). Note that in case of gridded uncertainty products, the random component becomes negligibly small, given long temporal and large spatial averages. Sampling uncertainties do not exist on an instantaneous basis and are therefore not considered in Eqs. (2)-(3).

Recall that $\sigma_{x,ran}$ in Eq. (3) represents the *overall* random uncertainty of $x$. To isolate its random satellite retrieval component ($E_{retr}^{ran}$), a random uncertainty decomposition is carried out, which is briefly reviewed in the following.

### 3.4  HOAPS-3.3 Random Uncertainty Decomposition

This section briefly summarizes the concept of random uncertainty decomposition. For more mathematical and technical details, the reader is referred to Kinzel et al. (2016).

Next to $E_{retr}^{ran}$, random uncertainty estimates resulting from collocations (e.g. black error bars in Fig. 1) include uncertainties associated with the collocation procedure ($E_C$) and *in situ* measurement noise ($E_{ins}$) (e.g. Bourras, 2006). To isolate $E_{retr}^{ran}$, multiple triple collocation (MTC) analysis is applied to matchups of $U$, $q_s$, and $q_a$ for the time period 1995–2008. MTC analysis includes a twofold triple collocation (TC introduced by Stoffelen, 1998), whereupon double collocated data described in Sect. 3.1 serves as input. Triplets incorporating two independent *in situ* measurements and one HOAPS-3.3 pixel represent

the first arrangement, whereas a single *in situ* record and two HOAPS-3.3 pixels of independent satellite instruments form the second triplet structure (see Fig. 1 in Kinzel et al. (2016)). The collocation criteria applied in Sect. 3.1 are adopted. Data poleward of 60° N/S is excluded to avoid biases associated with sea ice effects.

Subsequent to a bias correction with respect to the *in situ* measurements, the variances of differences between two independent data sources $X$ and $Y$, that is $V_{XY}$, are calculated following O'Carroll et al. (2008). Given three data sources and two types

of TCs, this results in six combinations of $V_{XY}$. Next, error models for both ship and satellite records are defined. In case of ship records, these include $E_{ins}$, whereas for satellite records, they incorporate satellite sensor noise ($E_N$, synthetically derived) and retrieval model uncertainty ($E_M$). Applying these error models to the derived $V_{XY}$ results in six equations incorporating $E_{ins}$, $E_M$, $E_N$, and $E_C$. These equations are successively solved for all random uncertainty sources as a function of the respective bulk parameter. $E_{retr}^{ran} = \sqrt{(E_M)^2 + (E_N)^2}$ is the pursued random satellite retrieval uncertainty.





Thus, MTC is a powerful tool to decompose *total* random uncertainties (i.e., $E_{sum} = E_{retr}^{ran} + E_{ins} + E_C$) inherent to $LHF$-related bulk parameters in order to isolate the random retrieval contribution $E_{retr}^{ran}$. Depending on the magnitude of the respective bulk parameter, the fractional contribution of $E_{retr}^{ran}$ to $E_{sum}$ is finally derived. That is, each entry of the total random uncertainty LUTs introduced in Sect. 3.2 is 'adjusted'. Table 1 presents a statistical summary of the instantaneous, decomposed random uncertainties inherent to $U$, $q_s$, and $q_a$.

## 3.5 Sampling Uncertainty

Next to systematic and random uncertainties, inhomogeneous sampling may occur, specifically when temporal and/or spatial resolution in observations are coarse. As remotely sensed data is measured at selected times only, spatial and temporal sampling uncertainties therefore become an issue (Gulev et al., 2010), as the diurnal cycle may not be captured correctly.

In a first step, daily mean sampling uncertainties of HOAPS-3.3 $LHF$-related parameters are derived, using high-resolution buoy measurements. Overall, data of eight tropical (PMEL, hourly resolution) and 15 extratropical (NDBC, 10-minute resolution) moored buoys account for a possible climate regime dependency. All chosen buoy records comprise several years of data and hardly show temporal data gaps. Here, the approach by Tomita and Kubota (2011) is followed to derive the sampling uncertainties by simulating satellite data overpasses based on the buoy records. In case of $U$ and $SST$, records are corrected for sensor heights and cool skin effects, respectively, as explained in Sect. 2.2. *In situ $LHF$* are computed by means of the COARE-2.6a algorithm (Fairall et al., 2003). Daily means of 'true' buoy data are derived by averaging all daily buoy records, where only high-quality data (indicated by quality flags 1–2) is considered. The weighted average of the two closest (in time) 'true' buoy observations to local satellite overpasses corresponds to the so-called 'simulated' satellite data record (Tomita and Kubota, 2011, their Fig. 2). All daily sampling uncertainties are derived as a function of the number of simultaneously operating SSM/I instruments. These daily values form the basis for the monthly averages of selected parameters, which are outlined in Table 2 (Sect. 4.3). The estimates are global means; an earlier, regime-dependent investigation resulted in negligible differences between the resulting sampling uncertainties.

## 4 Results and Discussion

### 4.1 Magnitudes of Decomposed Random Uncertainties

Table 1 presents a statistical summary of the instantaneous random uncertainty decomposition for the bulk parameters $U$, $q_s$, and $q_a$, following the approaches described in Sect. 3.2 and 3.4. Note that $E_N$ is not included, as its synthetically derived value (for procedure, see Kinzel et al., 2016) remains constant throughout the respective parameter range. Asterisked values indicate global mean weighted averages and pooled variances of Kent and Berry (2005), resulting from a semivariogram approach. These are based on their Fig. 1, taking the illustrated grid averaged random uncertainties, the standard deviation as well as the number of observations into account. In the following, individual contributions to the overall random uncertainties are discussed, but not shown in terms of supplementary figures.





$E_{retr}^{ran}(q_a)$ ranges between 0.3 and 1.8 g kg$^{-1}$, where minima (maxima) are found in Arctic (subtropical) $q_a$ regimes. Whereas largest relative uncertainties are associated with polar $q_a$ values, lowest relative contributions below 10 % are confined to the inner tropics. On average, both $E_c(q_a)$ and $E_{ins}(q_a)$ are approximately half the size of $E_{retr}^{ran}(q_a)$. The average of $E_{ins}(q_a)$ is 0.4 g kg$^{-1}$ below the mean estimate of Kent and Berry (2005). It is hypothesized that the lower estimate of $E_{ins}(q_a)$ is a direct

consequence of the rigorous *in situ* filtering procedure prior to MTC analysis. The difference may furthermore be triggered by the fact that Kent and Berry (2005) include data records dating back to the 1970s and 1980s, which may imply that ship records are included which do not fulfill the here applied quality control standards. In contrast to $E_{retr}^{ran}(q_a)$, $E_{ins}(q_a)$ increases rather linearly with $q_a$, which implies that smallest (largest) random *in situ* measurement uncertainties are found for lowest (highest) $q_a$. In contrast, $E_c(q_a)$ shows a similar distribution as $E_{retr}^{ran}(q_a)$, yet with considerably smaller amplitude. These

random collocation uncertainties range between 0.4 and 0.7 g kg$^{-1}$, corresponding to 3–18 %. A graphical illustration of the $q_a$ random uncertainty decomposition is shown in Kinzel et al. (2016) (their Fig. 2).

In case of $U$, all random uncertainties tend to be larger compared to $q_a$ in a relative sense. In contrast to $q_a$, all three relative uncertainties exhibit a clear increase over large ranges of $U$, where minima and maxima in $E_{retr}^{ran}(U)$ ($E_{ins}(U)$, $E_c(U)$) range between 1.0–2.6 m s$^{-1}$ (1.5–2.3 m s$^{-1}$, 0.8–2.0 m s$^{-1}$). Whereas $E_{retr}^{ran}(U)$ and $E_{ins}(U)$ are fairly constant for moderate wind

speeds before continuously increasing, $E_c(U)$ seems to already saturate for mean wind speeds in the order of 10 m s$^{-1}$ (not shown). Similar to $E_{ins}(q_a)$, the $E_{ins}(U)$ estimate of Kent and Berry (2005) is roughly 40 % larger. Again, this difference is suspected to arise from the differences in the data set compositions. Kent and Berry (2005) furthermore elucidate that no corrections for height or adjustments to the Beaufort scale have been applied to their data, which would have caused a reduction in random uncertainty of 13 ± 1 %, according to the authors. Yet, $E_{ins}(U)$ almost exclusively represents the largest contribution

to the random uncertainty budget of $U$. For all random uncertainty sources, strong wind regimes are linked to smallest relative uncertainties in the order of 12–15 %. In low-wind regimes, however, relative uncertainties exceed 50 % to even 100 %.

Both absolute and relative contributions of $q_s$-related random uncertainties remain well below those of $q_a$. Global mean values of all three random uncertainty sources are in the order of 0.5–0.6 g kg$^{-1}$. Regarding $E_{retr}^{ran}(q_s)$, this is comparable to the value published in e.g. McClain (1989), who estimated the global RMSE of AVHRR-derived $SST$ to be in the order of

0.6–0.7 K ($\widehat{=}$ 0.4–0.5 g kg$^{-1}$). Similar to $E_{retr}^{ran}(U)$, $E_{retr}^{ran}(q_s)$ ($E_{ins}(q_s)$) shows a positive proportionality with largest values of 0.9 g kg$^{-1}$ (1.5 g kg$^{-1}$). As for $E_{ins}(U)$, $E_{ins}(q_s)$ exceeds $E_{retr}^{ran}(q_s)$, specifically for $q_s$ larger than 8 g kg$^{-1}$. In contrast to $q_a$, relative uncertainties are smallest in extratropical regimes with contributions of merely few percent. Largest relative uncertainties remain well below those of $q_a$ and are in the order of 8–14 %.

## 4.2  Patterns of Random Retrieval Uncertainties

The results shown in Sect. 4.1 are expanded by showing the global patterns of $E_{retr}^{ran}$ in two-dimensional space.

Depending on the time period and thus on the number of SSM/I and SSMIS instruments in operation, the monthly global mean sum of instantaneous observations per 0.5°x0.5° grid cell ranges from approximately 90 (1988) to 650 (2006). In consequence, monthly means of $E_{retr}^{ran}$ are considerably below the systematic counterpart (see scaling effect of $N$ in Eq. (3)). Specifically from 1991 onwards, monthly globally averaged $E_{retr}^{ran}$ of $LHF$-related parameters only reach 0.5–3 %. This re-





duction becomes even more striking when investigating multi-annual or even climatological means; $LHF$-related $E_{retr}^{ran}$ virtually vanish on these scales. An increase (decrease) in these climatological random uncertainty values often directly results from a decrease (increase) in the number of pixel-level observations and thus not from a physical change due to shifts in the climate. This implies that results of trend analyses in random uncertainties, for example, may be misinterpreted. Therefore, the

attention is drawn to the pixel-level (*instantaneous*) random uncertainty fields, which are subsequently related to the systematic counterpart in terms of distribution and magnitude. This instantaneous point of view causes their orders of magnitude to be similar to the results of $E_{retr}^{ran}$ presented in Table 1. Note that the global averages shown in Fig. 2 in form of text strings are cosine-weighted, whereas the means illustrated in Table 1 do not take a regional dependency into account.

Figure 2 shows the instantaneous $E_{retr}^{ran}$ patterns of HOAPS-3.3 $LHF$-related parameters between 1988 and 2012. To a great

extent, Fig. 2a can be interpreted as a two-dimensional representation of the error bar magnitudes shown in Fig. 1a. Recall that the random uncertainties illustrated in Fig. 1a have not yet been corrected for the impact of $E_{ins}(q_a)$ and $E_c(q_a)$ (Sect. 3.4), which is why their magnitudes exceed those shown in Fig. 2a. Maxima above 1.5 g kg$^{-1}$ are located over all subtropical ocean basins, where $q_a$ is in the order of 13–17 g kg$^{-1}$. A reduction within the inner tropics is clearly resolved, specifically over the warm pool region. $E_{retr}^{ran}(q_a)$ sharply decreases poleward to values of 0.6–0.9 g kg$^{-1}$. The global mean instantaneous $E_{retr}^{ran}(q_a)$

takes on a value of 1.2 g kg$^{-1}$.

The distribution of instantaneous $E_{retr}^{ran}(U)$ (Fig. 2b) shows a rather reversed pattern of $q_a$ and closely resembles the climatological distribution of $U$ itself. The global mean is given by 1.0 m s$^{-1}$. Global maxima cover large areas of the extratropical oceans, specifically over the Southern Ocean. Here, averages partly exceed 1.5 m s$^{-1}$. However, this results in less than 15 % retrieval uncertainty in a relative sense (not shown). In contrast, instantaneous $E_{retr}^{ran})(U)$ remain low (that is, below 0.8 m

s$^{-1}$) over the (sub-) tropical ocean basins. This also applies to the warm pool area, which indicates a maximum in relative contribution close to 20 % due to climatological low wind speeds (not shown).

The pattern of instantaneous $E_{retr}^{ran}(q_s)$ (Fig. 2c) resembles that of $q_a$. However, the global mean magnitude of 0.3 g kg$^{-1}$ represents merely 25 % of the atmospheric counterpart. Absolute maxima in the order of 0.4 g kg$^{-1}$ are located over the Indo-Pacific warm pool region, which stands in contrast to the local minimum in that region for $q_a$. The comparatively small

$E_{retr}^{ran}(q_s)$ also find expression in the low global mean relative uncertainty of 2 % (not shown). Values exceeding 4 % are confined to the extratropical ocean basins on both hemispheres.

Instantaneous $E_{retr}^{ran}(LHF)$ (Fig. 2d) show a strong proportionality to the climatological mean $LHF$ pattern. In that respect, maxima are generally located over the subtropical central parts of all ocean basins (specifically the Indian Ocean) as well as along the western boundary currents. Respective values partly exceed 50 W m$^{-2}$. Apart from extratropical minima, low values

in the tropics are confined to the eastern margins of the basins and the warm pool region.

Figure 2e shows the instantaneous random uncertainty of $LHF$ relative to its natural variability. This variability has been defined as the pixelwise difference between the 5th and 95th percentile of instantaneous $LHF$ observations between 2000–2008, based on the F13 platform only. Globally averaged, the relative random uncertainty equals to 17 %. Due to the large range of $LHF$ along the western boundary currents (WBCs) and over the Central Indian Ocean, the absolute maxima seen in Fig. 2d





are not resolved in Fig. 2e. Largest relative uncertainties exceeding 25 % are confined to the Southern Central Tropical Pacific and along the equatorial Atlantic.

### 4.3 Monthly Mean Sampling Uncertainties

Table 2 summarizes the monthly mean sampling uncertainties of several $LHF$-related HOAPS-3.3 parameters as a function of concurrently operating SSM/I instruments. $SST$-related parameters show largest sampling uncertainties when three SSM/I instruments are simultaneously operating. This is not contradictory, as HOAPS-3.3 $SST$ are AVHRR-based and thus not linked to the coverage of SSM/I instruments. From a climatological perspective, all magnitudes are negligibly small compared to respective systematic uncertainties. Regarding the main bulk parameters, orders of magnitude closely resemble those of monthly mean scaled $E_{retr}^{ran}$. It is concluded that their relative contribution to the monthly mean uncertainty budget is in the order of merely 1–2 %. However, one should keep in mind that sampling uncertainties become essential on considerably shorter time scales, i.e., in the framework of (sub-) daily analyses.

### 4.4 Climate Means of HOAPS-3.3 Total Uncertainties

Figure 3a-e shows the distribution of the climatological *total* uncertainties ($E_{clim}$) between 1988 and 2012 for $LHF$ and its related bulk parameters. As the contribution of $E_{retr}^{ran}$ and sampling uncertainties converges towards 0% due to the vast number of observations, Figure 3a-e can also be treated as the systematic uncertainty distribution.

In an absolute sense, Fig. 3a mirrors the bias distribution shown in Fig. 1a. $E_{clim}(q_a)$ (Fig. 3a) generally range between 0.4–0.9 g kg$^{-1}$, where the global mean of 0.63 g kg$^{-1}$ is approximately half the size of the instantaneous random counterpart shown in Fig. 2a. Maxima are found over the tropical central and western Pacific Ocean as well as the Caribbean and off the easternmost tip of South America. In the framework of a $LHF$ intercomparison study, Smith et al. (2011) argue that satellite products have difficulties estimating $q_a$ due to persistent stratus clouds, as observed west of Peru over the tropical eastern Pacific. This conclusion may be the cause for the elevated systematic uncertainties over the tropical eastern Pacific. In contrast, minima are located along both extratropical belts poleward of 50–60° N/S. Secondly, isolated minima lie over the subtropical eastern margins of all ocean basins in the vicinity of 15–30° N/S, specifically over the Pacific basin. Interestingly, regions of comparatively low systematic uncertainties often coincide with regional maxima in random uncertainties (compare Fig. 2a). According to Fig. 1a, biases are smallest for climatological mean $q_a$ of 4–5 g kg$^{-1}$ and 13 g kg$^{-1}$, which fits well to the mentioned minima in Fig. 3a. Likewise, absolute bias maxima for $q_a$ of 10 g kg$^{-1}$ and 16–17 g kg$^{-1}$ are resolved in both Fig. 1a and Fig. 3a.

$E_{clim}(U)$ is shown in Fig. 3b. Its global mean equals to 0.81 m s$^{-1}$. On the one hand, maxima exceeding 1 m s$^{-1}$ are located along the extratropical storm tracks, specifically over the northern hemisphere. On the other hand, local maxima are found along broad regions at 30° S and further equatorward over the Central Indian Ocean, off the Arabian Peninsula (both monsoon-related), and the central Northern Tropical Pacific. With the exception of the Southern Ocean, this is in line with Brunke et al. (2011), who conclude that reanalysis -, satellite -, and combined data sets tend to overestimate wind speeds with respect to direct eddy covariance measurements, specifically over strong wind regimes. Monsoon-related characteristic features of Indian



Ocean $LHF$ variability, which also exhibit an impact on climatological uncertainties, are elucidated in e.g. Mohanty et al. (1996). Minima in the order of 0.5 m s$^{-1}$ are mostly confined to the eastern margins of all ocean basins (Fig. 3b). The maxima over the northern hemispheric storm track are associated with climatological mean wind speeds of 9–11 m s$^{-1}$. This range also reveals largest positive biases in the one-dimensional bias consideration with respect to the *in situ* source (analogously to Fig.

1, but not shown for $U$). This also targets the maximum over the central Northern Tropical Pacific and all southern hemispheric maxima along 40–50° S. Although climatological mean wind speeds maximise over the Southern Ocean, respective systematic uncertainties rather show a slight poleward decrease. Again, this is in line with results from the one-dimensional $dU$ analysis (not shown), which indicates that systematic uncertainties reduce for wind speeds above 12 m s$^{-1}$. Likewise, absolute bias minima are associated with low wind regimes in the order of 4–6 m s$^{-1}$. Climatologically lowest wind speeds of 2–4 m s$^{-1}$ are

for example found along the Pacific coast of Central America (15° N), over the Arabian Sea, and over the Indo-Pacific warm pool region. HOAPS-3.3 tends to underestimate these wind speeds, as is mirrored in moderate $E_{clim}(U)$ (Fig. 3b).

The climatological uncertainty estimates exceed those found in e.g. scatterometer records in comparison to buoy measurements (e.g. Verhoef et al., 2017). On the one hand, this is linked to the fact that estimates in Fig. 3b should be treated as upper-boundary uncertainty estimates. On the other hand, scatterometers are specifically designed to derive near-surface wind

speeds at highest accuracy. Passive microwave measurements, in return, allow for a much broader range of applications, which is a unique feature of HOAPS. An inclusion of scatterometer data into the HOAPS wind speed retrieval was not envisaged, due to differing overflight times and data coverage, i.e., additional uncertainties of unknown magnitude. Further potential uncertainty sources, which may contribute to the distribution shown in Fig. 3b, target currents, sea states, and the treatment of air mass density (i.e., the concept of stress-equivalent wind speeds, e.g. de Kloe et al., 2017).

$E_{clim}(q_s)$ covers the range of 0.1-0.6 g kg$^{-1}$ and its global average is given by 0.23 g kg$^{-1}$ (Fig. 3c). The pattern reflects a latitudinal dependency, which is equivalent to smallest (largest) biases towards the poles ((sub-) tropics). This observation is not generally valid, as is shown by the comparatively low values over large parts of the Eastern Tropical Pacific and Atlantic. Distinct maxima are found over the Arabian Sea and along northwestern Australia, the Caribbean, and west of Madagascar. Narrow bands of elevated systematic uncertainty are also resolved along the WBCs. With the exception of the WBCs, the

regions of maxima are exposed to $q_s$ in the range of 20–22 g kg$^{-1}$.

Figure 3d shows the resulting $E_{clim}(LHF)$. It closely resembles that of the global mean $LHF$ pattern itself with values ranging between roughly 15–50 W m$^{-2}$ and a global mean of 25.1 W m$^{-2}$. Relating this pattern to Fig. 3a-c shows a substantial contribution of $E_{clim}(q_a)$ to the absolute maximum of $E_{clim}(LHF)$ in the Northern/Southern Tropical Central Pacific, the Caribbean, and the western tropical South Atlantic (compare Fig. 3a). However, due to the large climatological mean $LHF$,

respective relative systematic uncertainties of $q_a$ are merely in the order of 5–7 %. Correspondingly, imprints of $E_{clim}(U)$ are clearly seen along the WBCs, the Central Indian Ocean (10–15 % in a relative sense), and off the Arabian Peninsula (partly exceeding 15 %) (Fig. 3b). Likewise, the maxima in $E_{clim}(LHF)$ over the Arabian Sea, along the northwestern coast of Australia, and close to Madagascar show the footprint of $E_{clim}(q_s)$ (Fig. 3c). However, relative systematic uncertainties in $q_s$ generally do not exceed 2.5 %. Locally, isolated $E_{clim}(LHF)$ maxima are resolved along 35° S. Specifically over the Agulhas

Current, Santorelli et al. (2011) conclude that different satellite data sets show discrepancies, as they are not able to properly



handle strong $LHF$ associated with storm systems and potential $LHF$ amplifications due to dry air advection northwards from the Antarctic (Grodsky et al., 2009). Furthermore, note that the maximum in the Arabian Sea is somewhat special, in as much as climatological mean $LHF$ in this region are elevated, yet not extraordinarily large. This striking uncertainty maximum may be linked to occasionally occuring advection of hot, dry air masses from the deserts, which poses problems to the HOAPS-3.3

satellite retrieval.

Figure 3e relates $E_{clim}(LHF)$ to its natural variability (compare Sect. 4.2). The global average is in the order of 12 %. Apart from the WBC regimes and the Southern Ocean, largest relative uncertainties are in line with the $E_{clim}(LHF)$ maxima illustrated in Fig. 3d.

### 4.5   Fractional contributions to total $LHF$ uncertainty

Simply comparing Fig. 3a-c to Fig. 3d allows for qualitatively assessing which $LHF$-related parameter contributes most to $E_{clim}(LHF)$. However, this does not permit a quantitative conclusion. Following a modified version of the 'Q-term' approach demonstrated in Bourras (2006), $E_{clim}(LHF)$ is decomposed into fractions associated with $U$, $q_s$, $q_a$, and $C_E$. Results indicate that the global mean contribution of $E_{clim}(q_a)$ is largest (60 %). This specifically targets the Central Northern and Southern Tropical Pacific, the Caribbean, the regime off the eastern tip of South America, as well as the Central Indian Ocean. On aver-

age, the contribution by $E_{clim}(U)$ takes on a value of 25 %. Local hotspots are considerably larger, especially over the Arabian Sea, along the WBCs, and off Northwestern Australia. The fractional contributions due to both $E_{clim}(q_s)$ and $E_{clim}(C_E)$ equal to 7.5 %, respectively. $E_{clim}(q_s)$ is largest over the Arabian Sea ($SST$ retrieval issues due to dust particles), whereas $E_{clim}(C_E)$ maximises over the Central Indian Ocean and along the North Atlantic WBC. The latter has also been shown by Bourassa et al. (2013), in as much as accuracy issues in $C_E$ tend to occur over very low and very high wind speed regimes.

All findings are in line with Bourras (2006), Liu and Curry (2006), Grodsky et al. (2009), and Santorelli et al. (2011), who conclude that the main $LHF$ uncertainty sources are related to the accuracy of $q_a$ (and $U$). Similar conclusions are drawn by e.g. Tomita and Kubota (2006), who show that the main source of discrepancy between tropical satellite and buoy estimates may be attributed to the accuracy of $q_a$. By comparison, HOAPS-3.3 uncertainty analyses are beneficial, as the findings of the above-quoted studies are restricted to either regional analyses, considerably shorter investigation periods, and/or comparatively

thin reference data bases.

### 4.6   Regional and Seasonal Analysis

Global mean $E_{clim}$ and $E_{retr}^{ran}$ of all $LHF$-related HOAPS-3.3 parameters are fairly constant in time throughout the whole climatology. During isolated time periods, however, absolute deviations from the global mean $LHF$ ($q_a$, $U$) uncertainty become as large as 18 % (3 %, 8 %).

Next to seasonal signals, these are footprints of distinct local anomalies. On the one hand, these anomalies seem to originate from events that temporarily modify the global climate. On the other hand, Figures 2-3 resolve considerable regional variability. Therefore, the aim is to (1) identify climate features that are manifested in both temporal and spatial uncertainty anomalies and





discuss their origin (descriptive only). At the same time, (2) regional uncertainty differences shall be highlighted by focusing on climate hotspots (Fig. 4a-c).

*Regarding (1)*: The imprints of moderate to strong El Niño events during boreal spring 1998 and 2010 are manifested in $LHF$-related $E_{clim}$ and $E_{retr}^{ran}$. During these events, wind speeds over the Pacific upwelling regime are 1.5–2.0 m s$^{-1}$ below the climatological average. As has been mentioned in Kinzel et al. (2016), this causes an increase in systematic uncertainties in $U$. Along with an enhanced $E_{clim}(q_s)$, the respective $E_{clim}(LHF)$ over the Pacific upwelling regime reaches 25 W m$^{-2}$ specifically during boreal spring 1998, which is approximately 10 W m$^{-2}$ above the seasonal mean and more than 50 % of climatological mean $LHF$. As $q_a$ are anomalously high with 20 g kg$^{-1}$, $E_{retr}^{ran}(q_a)$ is up to 0.2 g kg$^{-1}$ below the seasonal mean (see Fig. 2 in Kinzel et al. (2016) for clarification).

By contrast, global minima in $E_{clim}(LHF)$ and $E_{retr}^{ran}(LHF)$ are confined to boreal autumn 1991, taking on a mean value of 20 W m$^{-2}$ (33 W m$^{-2}$), respectively. These estimates are 20 % (11 %) below their climatological averages and are associated with absolute minima in HOAPS-3.3 $LHF$. The comparatively small systematic component is induced by $E_{clim}(U)$ ($E_{clim}(q_s)$) of -8 % (-14 %). The absolute minimum in $LHF$ and its uncertainties during 1991 is a footprint of the Mount Pinatubo eruption, which caused low-biased $SST$ due to AVHRR aerosol issues and thus unrealistically low near-surface humidity gradients (Romanova et al., 2010). Amongst others, this shortcoming in the HOAPS-3.3 climatology has already been picked up by Andersson et al. (2011).

*Regarding (2)*: Figures 4a-c summarize the ranges of seasonal, regime-dependent uncertainty distributions. The color-coded boxes in Figures 4a-c represent the expected parameter ranges when considering the systematic uncertainty contributions ($E_{clim}$). At the same time, the error bars indicate the instantaneous random uncertainty components ($E_{retr}^{ran}$). Both are shown separately, as they are independent of each other. With few exceptions, the random uncertainty contributions exceed the systematic counterpart, as is also mirrored in Figures 2e and 3e.

Figure 4a indicates that the total (i.e., $E_{clim} + E_{retr}^{ran}$) uncertainty ranges in $q_a$ are largest in (sub-) tropical regimes, concurrent to high $q_a$. In contrast to the Pacific upwelling region (red) and the Southern Ocean (cyan), the seasonal $q_a$ variability over the Indian monsoon regime (green), the North Atlantic basin (dark blue), and specifically the North Atlantic western boundary current (brown) is striking. This also finds expression in differences in absolute uncertainties of up to ±0.6 g kg$^{-1}$ between January and July. Largest uncertainties are in the order of ±2.40 g kg$^{-1}$ and are confined to the Indian summer monsoon season, whereas smallest uncertainties around ±1 g kg$^{-1}$ occur over the Southern Ocean.

Climatological regional wind speeds range between 4.5–11 m s$^{-1}$ (Fig. 4b). As for $q_a$, the seasonality is most pronounced over the Indian monsoon region, WBC, and the North Atlantic. Largest total uncertainties exceeding ±2 m s$^{-1}$ throughout the year are observed over the Southern Ocean, which is primarily due to large $E_{retr}^{ran}(U)$ (compare Fig. 2b). The Indian monsoon region is somewhat special, in as much as summertime total uncertainties are largest on a global scale, while wintertime ranges are almost 50 % lower.

Figure 4c presents regionally dependent $LHF$ and associated uncertainty ranges. As for Fig. 4a-b, seasonality is most distinct over the North Atlantic, WBC, and the Indian monsoon region. Largest $E_{clim}(LHF)$ exceeding ±35 W m$^{-2}$ are confined to the WBC regime (specifically during winter) and the monsoon region (climatological average, compare also Fig.





3d). Total uncertainty ranges maximise along the WBC, where $\pm65$–95 W m$^{-2}$ are to be expected, which is 2–3 times larger compared to the ranges observed along the Pacific upwelling regime. Grodsky et al. (2009), for example, recall that the Gulf Stream region is challenging due to strong surface currents and $SST$ gradients as well as intraseasonal dependencies of how the stratified atmospheric boundary layer amplifies air-sea interactions. This reasoning may also apply to the Agulhas and Kuroshio

region. The wintertime WBC uncertainty maximum is particularly caused by vast $E_{retr}^{ran}(LHF)$ of up to $\pm60$ W m$^{-2}$ (see also signal in Fig. 2d). By contrast, regional $E_{clim}(LHF)$ become largest in the Indian monsoon region, where their climatological average is in the order of $\pm40$ W m$^{-2}$ (compare also Fig. 3d).

## 4.7 Uncertainty Application: Trends in HOAPS-3.3 $LHF$

Figure 5 shows the HOAPS-3.3 global monthly mean $LHF$ (thin black line) between 1988-2012 (70° S-70° N, cosine-

weighted average). The global minimum below 80 W m$^{-2}$ during boreal summer 1991 is linked to the Mount Pinatubo eruption. Overall maxima in the order of 110 W m$^{-2}$ occur during 2008 and 2009.

    The bold black line in Fig. 5 shows the annual running mean climatology of HOAPS-3.3 $LHF$. On average, it increases by roughly 4.5 W m$^{-2}$ (4.7%) per decade (dark red line). If uncertainty ranges were discarded, this trend would be considered as significant at the 95 % level (p<0.00001, based on a two-tailed t-test). The addressed uncertainty estimates are illustrated as

grey shadings and represent $\pm1$ standard deviation of the 12-month running mean climatological uncertainty (global average). They take on a mean value of $\pm$ 17 W m$^{-2}$.

    A Bayesian approach to linear regression is applied including $LHF$ uncertainty estimates following Kelly (2007), which yields a large range of linear trends (light red lines). Although the majority has a positive slope, some even indicate a climatological decrease in $LHF$. In light of the illustrated uncertainty range, the mean upward trend in HOAPS-3.3 $LHF$ (dark red

line) should therefore be treated with caution, as the magnitude of linear increase lies well within the grey shaded area.

    The overall increase in $LHF$ has been elucidated in several studies concerning various $LHF$ data sets. Amongst others, it was already detected by Liu and Curry (2006) for HOAPS2 (Fennig et al., 2006), GSSTF2 (Chou et al., 2004), and reanalysis data (NCEP-R2, ERA-40; Kanamitsu et al., 2002; Uppala et al., 2005) between 1989–2000, specifically over the (sub-) tropics. The authors attribute it to increases in both $q_s$ and $U$, whereas the latter may be linked to stronger Hadley and Walker

Circulations (Cess and Udelhofen, 2003). Likewise, Gao et al. (2013) attribute largest contributions to observed positive trends in GSSTF2c $LHF$ to $q_s$ and $U$. Similar conclusions are drawn by Rahul and Gnanaseelan (2013) for the Indian Ocean, although local $LHF$ decreases in the Objectively Analyzed Air-Sea Heat Fluxes (OAFlux, Yu and Weller (2007)) are in line with findings from a model study by Held and Soden (2006). Yu and Weller (2007) present results from an OAFlux analysis and highlight the concurrent rapid warming of global SST (e.g. Levitus et al., 2005) and associated increasing $q_s$, especially over

the North Atlantic. Concurrently, $q_a$ decrease over the eastern Pacific and high southern latitudes, which adds to an increase in $LHF$. Mostly, the impact of increasing SST outperforms the positive trend in $q_a$, that is $\Delta q$ generally becomes larger. The authors also point out that trends in near-surface wind speeds are predominantly positive, which is in line with larger $LHF$. The global positive $LHF$ trend in the OAFlux product is strongest during the 1990s and is specifically evident along the WBCs, the Indo-Pacific warm pool region, and the Tropical Indian Ocean. The global mean increase of 9 W m$^{-2}$ between 1981 and



2002 is in the order of 10 %, which is in line with the findings illustrated in Fig. 5, yet one order of magnitude larger compared to the model study of Pierce et al. (2006). Santorelli et al. (2011) confirm this global mean $LHF$ increase in OAFlux and draw same conclusions for IFREMER $LHF$ (Bentamy et al., 2008), specifically for the North Atlantic. The increase in HOAPS-3 $LHF$ is also seen over the Southern Ocean, as has been investigated by Yu et al. (2011) due to increases in both $U$ and $\Delta$q.

Locally, these increases between 1988–2000 are in the order of 30 W m$^{-2}$.

Figure 5 also shows that recent global means decrease again. Time series analyses for single satellite instruments suggest that this is a physical signal (i.e., associated with either multi-annual variability or a climate signal), rather than being associated with intercalibration issues among SSM/I and SSMIS instruments. However, its decrease may be attributed to the slight negative $SST$ bias from 2011 onwards. This bias is caused by anomalously high NOAA-19 sensor noises, which themselves may be

traced back to erroneous flag assignments during cloud detection. This is thought to cause up to 5-10 % reduction in $LHF$. Closer investigations that involve other $LHF$ climatologies exceed the scope of this study, but are needed to interpret this gradual decay.

First intercomparisons of HOAPS-3.3 $LHF$ to *in situ* and further satellite climatologies have been carried out, where preliminary results indicate that nearly all compared data sets lie within the uncertainty range presented in Fig. 5 (not shown).

A more detailed intercomparison study is envisaged; it will benefit from uncertainty estimates available in NOCSv2.0 and allow for concluding whether global mean deviations among the data sets lie within or outside of the HOAPS-3.3 prescribed uncertainty range.

## 5   Conclusions and Outlook

By means of multi-dimensional bias and MTC analysis, a universal approach for characterizing systematic, random retrieval,

and sampling uncertainties inherent to HOAPS-3.3 $LHF$-related parameters has been presented. HOAPS-3.3 can therefore be considered as the first $LHF$ satellite-only climatology including instantaneous and gridded uncertainty estimates. It has been shown that maxima of systematic uncertainties ($E_{clim}$) reach up 50 W m$^{-2}$, specifically over the large regions of the subtropical oceans (mainly $q_a$-induced) and along the western boundary currents (mainly $U$-induced). Instantaneous random retrieval uncertainties ($E_{retr}^{ran}$) maximise along 20–30° N/S with values up to 60 W m$^{-2}$, clearly showing the footprint of random

uncertainties of $q_a$. From a climatological perspective, all random retrieval uncertainty components contribute to the total uncertainty by merely 1–2 % on a monthly basis (and even less for longer periods), which also accounts for respective sampling uncertainties. Considerable regional and seasonal variability of $LHF$ uncertainty ranges have been resolved from an instantaneous point of view, with maxima over the Gulf Stream and Indian monsoon region during boreal winter. Climate events, such as strong El Niño signals and the Mount Pinatubo eruption, are well manifested in both systematic and random $LHF$

uncertainties, even on a global scale. In light of the available uncertainty estimates, it has been shown that the positive trend in global mean $LHF$ during the last 25 years lies within the derived uncertainty boundaries.

A new version of HOAPS-3.3, that is HOAPS-4.0, will be released in mid 2017. Major changes compared to HOAPS-3.3 include a temporal extension up to 2014, a new $SST$ product (Version 2 of the NOAA Optimum Interpolation $SST$ (OISST)





product, Reynolds et al. (2007)), and the implementation of a 1D-Var retrieval for several geophysical parameters. Preliminary results suggest that the new $U$ estimates have improved compared to HOAPS-3.3 in terms of bias and RMSD behaviour relative to *in situ* ground reference data. In consequence, estimates of $LHF$ and $E$ will be updated, along with $LHF$-related uncertainty estimates.

Results of the Q-term analysis presented in Sect. 4.5 and other studies suggest that more effort is necessary to improve the $q_a$ retrieval. This would ultimately reduce the overall $LHF$ uncertainty, which, according to e.g. Bourras (2006), ought to be below 10 W m$^{-2}$ for a quantitative use over the global oceans. In the framework of the HOAPS-4.0 release, this value has also been declared as the target requirement for the global mean $LHF$. An increase in the reliability of HOAPS-3.3 $LHF$-related parameters could for example be achieved by referring to a new ground truth reference. Freeman et al. (2016), for example,

recently presented a new version of ICOADS (release 3.0, up to 2014), highlighting its improvements compared to earlier versions, which target topics such as data quality, data traceability, and data base extension. Apart from new *in situ* reference data, the effect of approximations in bulk flux parameterizations should also be picked up, as has been done in detail in Brodeau et al. (2017). Amongst others, this concerns implications of sensor height corrections, algorithm choices, the $q_s$ reduction due to the salinity effect, cool skin/ warm layer effects, and the assumption of constant sea level pressure.

According to Andersson et al. (2011), the E-P budget of HOAPS-3.2 is not closed. This also accounts for HOAPS-3.3, with a climatological mean value of 0.45 mm d$^{-1}$ (1988–2012, 70° S-70° N). Long-term run-off estimates are summarized and published by the Global Runoff Data Center (GRDC), adding up to a mean value of 0.34 mm d$^{-1}$ (Wilkinson et al., 2014). According to Andersson et al. (2011), the uncertainty of these run-off estimates is in the order of 10–20 %. Comparing these values to the HOAPS-3.3 global freshwater flux leaves an imbalance of approximately 0.10 mm d$^{-1}$, which is 0.30 mm d$^{-1}$

below the HOAPS-3.2 estimate and can be evaluated as an improvement towards closing the global freshwater flux imbalance. As $E_{clim}(E)$ is in the order of $\pm$ 0.6 mm d$^{-1}$, the imbalance clearly lies in the range of freshwater flux uncertainty.

    Recall, however, that uncertainty estimates of HOAPS-3.3 precipitation have not been accounted for in this quantitative estimation. Generally, the availability of remotely sensed precipitation uncertainty estimates is complicated by sparse reference data and its intermittency. Tian and Peters-Lidard (2010), for example, have taken on the challenge of creating global maps of

uncertainties in satellite-based (i.e., six TRMM-era data sets) precipitation measurements. In conclusion, overall uncertainties range between 40–60 % over the tropical oceans, whereas uncertainties may exceed 100 % over the higher-latitudinal regimes poleward of 40° N/S. A recent study by Burdanowitz et al. (2016) presents an automatic phase distinction algorithm for optical disdrometer data. Together with a continuously growing high-quality *in situ* data base of ship-based precipitation measurements (OceanRAIN, Klepp (2015)), it will serve as a valuable basis for a characterization of HOAPS-3.3 precipitation and hence

freshwater flux uncertainty ranges in the near future. Accuracy assessments of global rainfall estimates can also be achieved by means of triple collocation analysis, as is demonstrated in Massari et al. (2017).

    Future work also aims at investigating trends in water vapour transports (WVT), using HOAPS-3.3 monthly mean freshwater fluxes. Sohn and Park (2010), for example, demonstrated that trends in WVT can be used to examine circulation changes and conclude that the large-scale Hadley Circulation has experienced an increase in strength since 1979. Similarly, Durack et al.

(2012) recently highlighted a considerable water cycle intensification during global warming. Available uncertainty estimates



will allow for quantifying the WVT uncertainty range, the necessity of which has been picked up by e.g. Sohn et al. (2004).

*Data availability:* HOAPS-3.3 is a prolongation of HOAPS-3.2 and is based on a pre-release of the CM SAF SSM/I and SSMIS

5  FCDR. It was created in the framework of the DFG FOR1740 research activity for internal use. The monthly mean HOAPS-3.2
climatology and the respective FCDR are publicly available and may be downloaded free of charge (http://www.cmsaf.eu/EN/
Products/DOI/Doi_node.html). Instantaneous and gridded HOAPS-3.3 data are available upon request from the author.

10  *Competing Interests:* The authors declare that they have no conflict of interest.

*Acknowledgements.* J. K. is funded by the German Science Foundation (DFG FOR1740/FOR21740). The funding for the development and
implementation of the collocation software was provided by the German Meteorological Service (DWD). HOAPS-3.3 was generated within
DFG FOR1740. DWD-ICOADS data was gratefully obtained from the Marine Climate Data Center (DWD).



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



**Table 1.** Absolute and relative random statistical measures resulting from the multi-dimensional LUTs and random uncertainty decomposition (Sect. 3.2, 3.4). 'stddev' = standard deviation, 'abs' = absolute, 'rel' = relative. Apart from the $LHF$-related bulk parameters themselves ($U$, $q_s$, and $q_a$), global mean ranges of the random retrieval- ($E_{retr}^{ran}$), random collocation- ($E_c$), and random *in situ* measurement uncertainty ($E_{ins}$) are shown. Relative measures result from bin-wise relative uncertainty calculations. For comparison, the asterisks indicate respective estimates published in Kent and Berry (2005), which are based on a semivariogram approach.

| parameter / stat. measure | mean | stddev | min (abs) | min (rel) | max (abs) | max (rel) |
|:---:|:---:|:---:|:---:|:---:|:---:|:---:|
| $q_a$ [g kg$^{-1}$] | 8.8 | 4.4 | 2.8 | / | 19.3 | / |
| $E_{retr}^{ran}(q_a)$ | 1.0 | 0.3 | 0.7 | 6 % | 1.8 | 24 % |
| $E_c(q_a)$ | 0.5 | 0.1 | 0.4 | 3 % | 0.7 | 18 % |
| $E_{ins}(q_a)$ | 0.5 [0.9*] | 0.3 [0.3*] | 0.1 | 4 % | 1.2 | 7 % |
| $U$ [m s$^{-1}$] | 7.9 | 3.6 | 1.8 | / | 15.4 | / |
| $E_{retr}^{ran}(U)$ | 1.4 | 0.4 | 1.0 | 12 % | 2.6 | 63 % |
| $E_c(U)$ | 1.4 | 0.3 | 0.8 | 12 % | 2.0 | 44 % |
| $E_{ins}(U)$ | 1.8 [2.5*] | 0.2 [0.4*] | 1.5 | 15 % | 2.3 | 111 % |
| $q_s$ [g kg$^{-1}$] | 10.2 | 5.7 | 4.5 | / | 24.3 | / |
| $E_{retr}^{ran}(q_s)$ | 0.5 | 0.2 | 0.2 | 2 % | 0.9 | 9 % |
| $E_c(q_s)$ | 0.5 | 0.1 | 0.4 | 2 % | 0.6 | 14 % |
| $E_{ins}(q_s)$ | 0.6 | 0.5 | < 0.1 | 1 % | 1.5 | 8 % |





**Table 2.** Monthly mean HOAPS-3.3 $LHF$-related sampling uncertainties as a function of simultaneously operating SSM/I instruments. $q_a$ = "hair", $U$ = "wind", $q_s$ = "hsea", $LHF$ = "late", $SST$ = "asst", $E$ = "evap", air temperature = "tair". All magnitudes are negligible compared to the instantaneous random ($E_{retr}^{ran}$) and climatological uncertainties ($E_{clim}$) presented in Sect. 4.2 and 4.4

| # of satellites / parameters | "hair" [g kg$^{-1}$] | "wind" [m s$^{-1}$] | "hsea" [g kg$^{-1}$] | "late" [W m$^{-2}$] | "asst" [K] | "evap" [mm d$^{-1}$] | "tair" [K] |
|---|---|---|---|---|---|---|---|
| 1 | 0.05 | 0.14 | 0.04 | 2.3 | 0.04 | 0.08 | 0.08 |
| 2 | 0.03 | 0.12 | 0.04 | 1.9 | 0.03 | 0.07 | 0.05 |
| 3 | 0.03 | 0.11 | 0.05 | 1.8 | 0.04 | 0.06 | 0.04 |



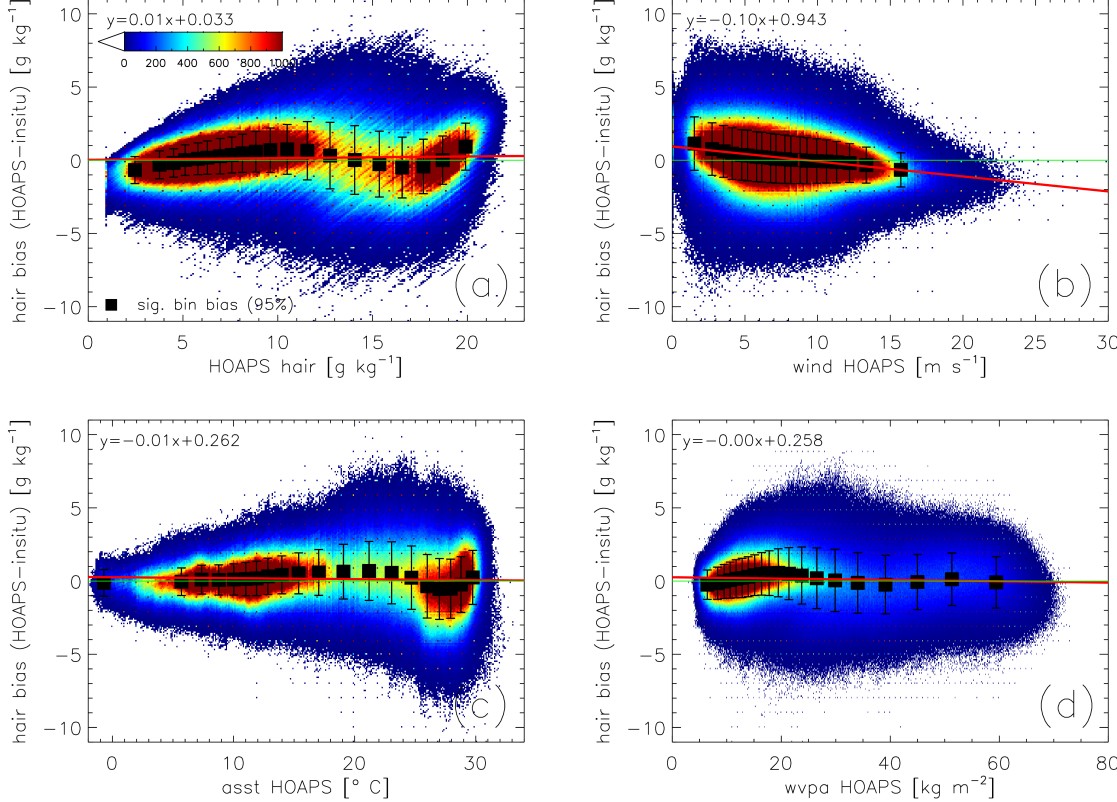

**Figure 1.** Scatter density plots of $q_a$ bias (HOAPS-3.3 minus *in situ*, g kg$^{-1}$) as a function of (a) $q_a$ ("hair"), (b) $U$ ("wind"), (c) $SST$ ("asst"), and (d) water vapour path ("wvpa"), based on global double collocations between 2001 and 2008. The black squares and error bars represent bin-averaged systematic uncertainties (significant at the 95 % level) and their standard deviations, whereby each bin contains 5 % of all double collocated matchups. Note that the bars include random uncertainty contributions by the satellite retrieval, the collocation procedure, and the *in situ* measurement uncertainty. (a) is a revised version of Fig. 3 published in Kinzel et al. (2016).



**Figure 2.** Temporal averages (1988-2012) of HOAPS-3.3 instantaneous $E_{retr}^{ran}$ of (a) $q_a$ ("hair"), (b) $U$ ("wind"), (c) $q_s$ ("hsea"), and (d) $LHF$ ("late"). (e) Relative random retrieval uncertainty of HOAPS-3.3 $LHF$ with respect to its natural variability. This variability is defined as the range between the 5th and 95th percentile of instantaneous $LHF$ between 2000-2008. The global averages (text strings) were derived by considering a latitudinal cosine-dependency. All patterns result from the multi-dimensional bias analyses, random uncertainty decompositions, and, in case of (d), uncertainty propagation described in Sect. 3.2-3.4.







**Figure 3.** Temporal averages (1988-2012) of HOAPS-3.3 climatological total uncertainties ($E_{clim}$) of (a) $q_a$ ("hair"), (b) $U$ ("wind"), (c) $q_s$ ("hsea"), and (d) $LHF$ ("late"). (e) Climatological mean relative $E_{clim}(LHF)$ with respect to its natural variability. This variability is defined as the range between the 5th and 95th percentile of instantaneous $LHF$ between 2000-2008. The global averages (text strings) were derived by considering a latitudinal cosine-dependency. All patterns result from the multi-dimensional bias analyses and subsequent uncertainty propagations described in Sect. 3.2-3.3.





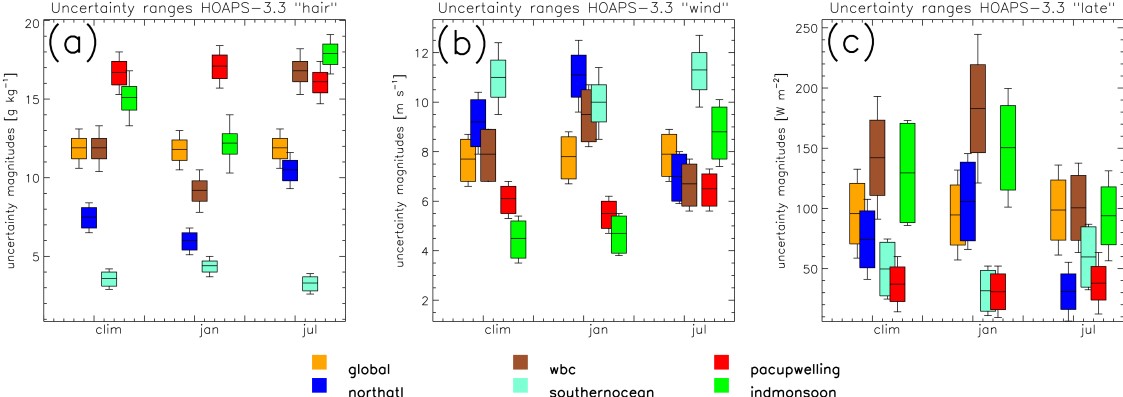

**Figure 4.** (a) Expected ranges of $q_a$ ("hair") as a function of different regions and seasons, while considering both $E_{clim}$ and $E_{retr}^{ran}$: global (orange), North Atlantic (60° W–5° E, 35–65° N, dark blue), North Atlantic Western boundary current (WBC, 60–80° W, 30–40° N, brown), Southern Ocean (50–60° S, cyan), Pacific upwelling regime (80–100° W, 5° N–5° S, red), and Indian Monsoon region (50–75° E, 15–30° N, green). Whereas the color–coded boxes show the expected systematic uncertainty, the bars indicate the random uncertainty component. (b) As for (a), but for $U$ ("wind"). (c) As for (a), but for $LHF$ ("late").



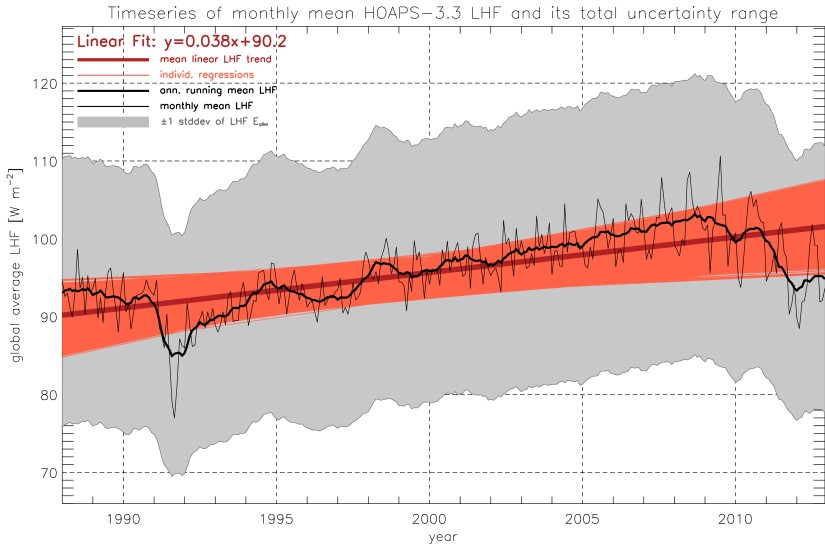

**Figure 5.** The thin (thick) black line shows the monthly (annual running mean) time series of HOAPS-3.3 $LHF$ (70° S-70° N, cosine-weighted average). The dark red line illustrates the linear trend, which takes on a value of 4.5 W m$^{-2}$ per decade (p<0.00001, based on a two-tailed t-test). The grey shading represents $\pm$ 1 standard deviation ("stddev") of the annual running mean $E_{clim}$. The light red regression lines were iteratively derived following Kelly (2007) by taking $\pm$ 1 stddev of $E_{clim}$ into account.