# Peer review of "Uncertainty Characterization of HOAPS-3.3 Latent Heat Flux Related Parameters"

_Atmospheric Measurement Techniques, 2017_

## Referee Comment (RC1) · Anonymous Referee #1 · 8 Sep 2017

Journal: AMT Title: Uncertainty Characterization of HOAPS-3.3 Latent Heat Flux Related Parameters Author(s): Julian Kinzel et al. MS No.: amt-2017-176 MS Type: Research article

Review:

(General comments) The authors investigated uncertainty characterization of HOAPS-3.3 latent heat flux (LHF) related parameters. Since latent heat fluxes are one of the main contributors to the global energy budget as they pointed out in their abstract, estimation of uncertainty of LHF is quite important, especially in climate studies. This article is based on Kinzel et al. (2016). However, the paper is not referred in the present introduction. It is curious. The purpose of this article is not so clear for me. I think the purpose of this study is comprehensive estimation of uncertainty character-

ization of HOAPS-3.3 latent heat flux (LHF) related parameters in addition to specific humidity examined in Kinzel et al. (2016). We can find the word of " inherent" in the title of Kinzel et al. (2016), but cannot the word in the tile of this article. I agree that it is quite welcome to be provided a data set such as HOAPS-3.3 with uncertainty estimates. However, we are interested in whether the estimated uncertainty is common in (satellite) products or inherent in HOAPS-3.3. If the present results are inherent in HOAPS-3.3, the results are useful for only people to use HOAPS-3.3. However, if the results are common in most satellite products, the value of this article is considerably larger. For example, the authors attribute the global minimum during boreal summer 1991 to the Mount Pinatubo eruption. However, we cannot find the minimum in 1991 in other products except HOAPS (Iwasaki and Kubota, 2014, Fig.6 (a)). Therefore, the minimum may be due to the HOAPS retrieval error related to the Mount Pinatubo eruption. Also, since all HOAPS parameters are derived from SSMI and SSMIS microwave radiometers, the sampling errors are expected to be large compared with other products using many kinds of microwave radiometers. As a result, we can easily expect that the uncertainties are different among each satellite product. If possible, we would like to know uncertainties about other products in order to judge whether the estimated uncertainty for HOPAS-3.3 in this study is common or not. I guess it is not so easy for the authors to estimate uncertainties for other products. If so, I would like the authors to investigate the relation between the uncertainties of HOAPS-3.3 obtained by this study and the differences between HOAPS and other products, pointed out by previous paper. Also some parts of the paper may be eliminated or reduced. For example, although the second paragraph in the section 5 introduces HOAPS-4.0, I feel the paragraph is not necessary in this section. Moreover, the authors discuss about precipitation in this section, but I think this issue may exceed the scope of this study because they do not carry out uncertainty estimates of HOAPS precipitation here. It is my opinion that the manuscript needs major revision before it can be accepted for publication. As far as the usefulness of the present results is limited in HOAPS-3.3ïïjŇit is difficult to accept for AMT publication.

ïijĹSpecific CommentsïijĽ P.1, L.1 " of LHF" −→ " of in situ LHF"

P.3, L.21-27 In this paragraph, we need clear description about characteristics related to uncertainties, of HOAPS LHF product compared with other products obtained by numerous intercomparison studies

P.5,L.16-21 Large El Nino and La Nina occurred in 1997-1998. Therefore, 1997-1998 is a special period. Why did the authors use the data in this period?

P.5,L.33-34 The assumption of a constant relative humidity of 80 % and air-sea temperature difference of 1 K is considerably artificial. To what extent does have the assumption impact on estimation of uncertainty?

P.6,L.25 (2003) —(2013)

P.8,L.15 ãĂĂIn what ways are these features similar?

P.8,L.22 " off the Arabian Peninsula" We cannot recognize the data off the Arabian Peninsula in Fig. 1. We need the distribution of average qa for this.

P.9 L.11 Is the bin width equal or not? How did you determine the bin width?

P.9, L.17 Why did you choose the different data period between (dqa, dU) and (dqs)?

P. 10, L.4 The average of daily coefficients is applied for estimation of instantaneous LHF uncertainties here. Why are not instantaneous values but daily values applied? Also, is the difference between daily and instantaneous coefficients small or large?

P.10,L.9 Could you explain about the definition of " gridded uncertainty products"?

P.10,L.17 What is a true value for Ec?

P.11,L.19-22 Here, all daily sampling uncertainties are derived as a function of the number. However, sampling error for a daily-mean value depends on not only the number but also observation times.

P.12,L.1-3 We find several geographical words such as " Arctic", " polar" and " inner

tropics". However, It is difficult for us to obtain the relation between the ranges of the random satellite retrieval uncertainty and the geographical location from Fig. 1 and Table 1. Also are the values shown in this paragraph consistent with those in Table 1? For example, " 0.3 and 1.8gkg-1" is " 0.7 and 1.8gkg-1" in line 1?

P.12, L. 1-28 Accuracy of in situ data is considerably different depending on used sensors. For example, the accuracy of wind speeds is 1.0m/s or 10% for usual NDBC buoys, while that is 0.3 m/s for TOA buoys. Are these differences between them negligible for the present analysis?

P.14, L. 13 Could you tell me the definition of the climatological total uncertainties (Eclim)? Are the climatological total uncertainties (Eclim) different from the systematic uncertainty?

P.16,L.28 What is the meaning of "isolated time periods"?

P.17,L. 3-19 Eclim is considered to be only one value from the meaning of a climatological value. Is it right? If so, I cannot understand the meaning of " respective Eclim over the Pacific upwelling regimes reaches 25 W m-2 specifically during boreal spring 1998" " found in line 6-7.

P.17, L.28 "climatological regional wind speeds range between 4.5.-11 m s-1 (fig.4b). As for qa" ——→ "climatological regional uncertainties in wind speeds range between 4.5.-11 m s-1(fig.4b). As for U"

P.18, L.10 The global minimum during boreal summer 19991 is linked to the Mount Pinatubo eruptions. However, the remarkable minimum can be found in only HOAPS product and cannot be found in other products as shown in Fig. 6(a) of Iwasaki and Kubota (2014). Therefore, the minimum would be related to retrieval model uncertainty. The present analysis can investigate this issue and present its effectiveness by the investigation.

P.18,L. 15 As mentioned before, could you please explain about definition of climatological uncertainty? I cannot catch the meaning of " the 12-month running mean climatological uncertainty". Is a climatological uncertainty defined each month?

P.18, L. 21-P.19,L. 5 In this paragraph, the results by many previous studies are introduced. However, the relation between the results and what Fig.5 shows is not so clear. I wonder this paragraph is necessary.

Fig 2. (c) and Fig. 3. (c) It is difficult to know the distribution pattern in these figures. How about the change of a color bar?

Please also note the supplement to this comment:
https://www.atmos-meas-tech-discuss.net/amt-2017-176/amt-2017-176-RC1-supplement.pdf
* * *

---

## Referee Comment (RC2) · Anonymous Referee #2 · 26 Oct 2017

Uncertainty estimates should be provided by every gridded product, although only a few products do. This study provides uncertainty estimates for the HOAPS2.2 latent heat flux and flux-related variables, which is certainly a positive step forward. The definition of uncertainties in this manuscript, however, is different from the definition commonly used in literature. It is not clear how instantaneous and climatological uncertainties are related to systematic, random, and sampling uncertainties. The use of "climatological uncertainties" is particularly atypical. In addition, the long-term upward trend of latent heat flux seems erroneous from the perspective of the global water budget.

Main comments are provided as follows.

Page 5, Line 10, ". . . the pixel-level HOAPS-3.3 data in sensor resolution is used. . .".

What are the spatial and temporal resolutions of the pixel-level HOAPS-3.3 data? Which nine sensors are used in the pixel-level HOAPS-3.3 climatology?

Page 5, Line 15: what is the temporal resolution of qa retrievals? And at what height?

Page 5, Line 32: which surface pressure data are used in computing LHF?

Page 5, Line 33: " . . . surface air temperature, which is estimated by assuming a constant relative humidity of 80 % (Liu et al., 1994) and air-sea temperature difference of 1K". How accurate is this assumption? During winter cold air outbreaks over the western boundary current regions, the air-sea temperature differences can exceed 10 K. In this case, the assumption will lead to a bias in air temperature. How is surface air temperature compared to the in situ dataset?

Page 6, Line 1: Provide a map showing the spatial distribution of in situ (ship and buoy) reference data density over the global domain.

Page 6, Lines 4-5: Does the reference dataset include the 1996-97 period that is used in training qa algorithm?

Page 7, Lines 28-29: The "instantaneous and climatological uncertainties" are not explained. How are they related to systematic, random, and sampling uncertainties?

Page 8, Line 10: Definition of water vapour path?

Page 8, Lines 11-14: It seems that HOAPS qa is wet biased in the tropical wet zone and dry biased in the subtropical dry zone. The bias pattern seems to be similar to GSSFT v3 qa product (Prytherch et al. 2014, Int. J. Climatol.; Jin et al. 2015, J. Atmos. Ocean. Technol.).

Page 8, Lines 21-22: Indeed, the 1-D bias analysis is not sufficient. Please provide a figure showing the global pattern of the mean differences between HOAPS and the reference data. Need to discuss the uncertainty pattern in terms of humidity regimes.

Page 8, Line 24: "Recall that the aim is to characterize uncertainty and not bias pat-

terns". The sentence is confusing. Bias is one kind of uncertainties.

Page 10, Eqs (2)-(3): Which figures are produced from Eqs.(2)-(3)?

Page 10, Line 10: Why only random satellite retrieval component, not the total random uncertainty, is computed?

Pages 10-11, sections 3.4-3.5: The two sections are not directly related to any figures. Suggest to revise and combine.

Page 13, Line 10: Fig.2 is regarded as a 2-D representation of the error bar magnitude of Fig.1a. A figure showing the global pattern of HOAPS3.3 - minus -in situ needs to be provided to help interpret Fig.2.

Page 13, Fig.2: The instantaneous random uncertainty map of qa (Fig.2a) has a pattern similar to the uncertainty map of qa produced by OAFlux (Yu et al. 2008, OAFlux technical report), though HOAPS3.3 has a much larger magnitude.

Page 14, Line 3: In addition to Table 2, please add a zonal-mean average of the monthly mean sampling uncertainties to show the latitudinal distribution of the uncertainties.

Page 14, Line 13: How is Eclim defined? Please provide a mathematical expression of Eclim.

Page 14, Line 15: "Figures 3a-e can also be treated as the systematic uncertainty distribution". What is the relation between Figures 3a-e and the mean difference map of HOAPS3.3 minus in situ? See comment Page 13, Line 10. The maps shown in Figures 3a-e are not bias patterns, as bias has both positive and negative signs. What is the meaning of the systematic uncertainty?

Page 18, Line 5: "On average, it increases by roughly 4.5 W m-2 (4.7%) per decade...". Which term gives rise to this large increase, qa – qs or U? The continuing increase in LHF during the "hiatus" period in the 2000s does not seem realistic from the perspective

of the global water budget balance (see Robertson et al. 2014, J.CLim).

Page 19, Line 14: Remove the sentence. Aren't the uncertainty estimates supposed to be a common practice for all gridded products?

---

## Referee Comment (RC3) · Anonymous Referee #3 · 8 Nov 2017

This paper develops a method to estimate errors of the HOAPS-3.3 ocean evaporation product. Ocean evaporation is a key component of the earth's energy and water balance, and error estimates are very much needed in budget studies and climate drift investigations. Too often conclusions are drawn on the basis of climatological data that does not have the accuracy that is needed. So this paper is very welcome as a complement to the HOAPS evaporation data. The methodology by itself is also of scientific interest because it can be applied to other data. The triple co-location method is used here in an interesting way as it allows to isolate the co-location error from the total error. Also random errors and systematic errors are distinguished, although I have questions about the analysis of systematic errors.

I have the feeling that the authors have done a lot of good work that is of interest to the

research community, and therefore I feel that the work should be published. However, the way the work is presented can and should be improved.

I have two major concerns:

1. The paper is hard to read. Often it requires re-reading a paragraph a number of times, to understand. It also has to do with the structure of the paper. It would help to define the main methodology of the data analysis and to have this as the main thread throughout the paper. I think I understand the methodology, but I am still not sure. Let me explain my interpretation of the method:

(i) Four dimensional look-up tables (LUT) are created of co-located data, so differences between data sets are stratified according to q_a, U, SST, and wvpa.

(ii) The mean difference between HOAPS and insitu data are interpreted as biases.

(iii) The variances of the difference are used for the triple co-location method, resulting in error estimates.

(iv) This results in LUT's of biases and random error estimates.

(v) In applications (e.g. global maps of mean and random error of q_a) the observations of q_a, U, SST and wvpa point to the table and provide errors of each observation. These can be averaged to obtain the desired map.

I feel that it would be helpful to describe upfront that this is the general methodology and follow it throughout the paper. So this would lead to 3 main sections in the paper: (i) Description of the methodology, (ii) Results of the methodology, i.e statistics on the LUT data, and (iii) Application to HOAPS evaporation. In case I am completely wrong on the interpretation of the paper, there is even more reason to be clear about the methodology.

Another question is: what is the main result of the paper? If my interpretation is correct, then the 4-dimensional table of error estimates is the main result, because it would

allow a user to make estimates of anything he/she is interested in (e.g. monthly averages, daily averages, or El-Nino years). So it is worth thinking about communicating this 4D table to the users. Most of the current paper is about applying the methodology, but these are in fact just examples.

2. Estimation of biases is non-trivial. In fact this is very important because, as the authors point out, for long term averages the systematic errors dominate. My concern is two-fold:

(i) I have the feeling that it is assumed that DWD-ICOADS data is bias-free? If this is the basis for the bias estimates, then it deserves more discussion also in view of what has been published in literature.

(ii) Fig. 1 is used as an example to illustrate the estimation of biases. However, it is likely that artificial biases occur in binned scatter plots of noisy data if correlated variables are used on abscissa and ordinate. This applies to Fig. 1a where hair(HOAPS) is used on both vertical and horizontal axes. It also applies to hair versus wind because these variables are correlated due to the physics of the mixing (more wind brings hair closer to the surface value). To check, one could e.g. bin the differences of Fig. 1a in classes of hair(insitu). Also hair(insitu) is noisy because it has large representativeness errors (point observation, whereas HOAPS has a large footprint).

Finally, if one can be confident about the bias estimation, then it should also be trivial to apply a bias correction to HOAPS. This would just leave the uncertainty in C_E which is a parametrization constant used for satellite as well as in-situ data. Please discuss.

More detailed comments:

Section 1

Although well written, the introduction is rather long and contains sometimes fluffy language. For instance, the first 24 lines illustrate the importance LHF, which is well known and can be much shorter. The second sentence is another example. A side

issue is introduced by referring to turbulent heat fluxes (not latent heat fluxes). A good overview of the literature is given, but no reference is made to an earlier study by Kinzel et al. on the estimation of uncertainty of q_a. It is important to point out in the introduction what is new compared to the earlier work. My interpretation is that Kinzel et al. (2016) does an error analysis on q_a, and that the current paper extends it to U, q_s and LHF. It is important to clarify this.

The introduction makes references to other data sets and to studies that provide error estimates. However, nothing is said about published error estimation methods. Since it is the topic of the current paper, it is necessary to explain what is different about the own method compared to others.

Page 5, Line 32

The sentence with "The latter depends" suggests that it refers to q_a in the sentence before, but what it intends to say is that the COARE algorithm needs stability and that specific assumptions are made. Please rephrase.

Page 6, Line 20-24

The non-correction of q_a for measuring height is confusing. Why not using the real measuring height in the bulk formula? Perhaps it is possible to say in one sentence what the results are of the height difference effects as estimated by Kent et al. (2014)

Page 7, Line 13

Cool skin corrections are applied to in situ observation but not to HOAPS-3.3 SST (AVHRR based). This makes sense in priciple because AVHRR measures the skin temperature. However, there must be a calibration procedure of AVHRR, which is probably against bulk SST data. So, what does calibrated AVHRR data represent, bulk or skin SST?

Pages 4-5 section 2.1

It would be informative to mention pixel size of the microwave sensors.

Page 8, Line 11

The sentence "Figure 1a overestimates ...." is confusing. Formally it is correct, but, after reading the first time it suggests that the biases range from 7-12 g/kg and that the plot is for the inner tropics.

Page 8, Line 17

The expression "over-(under-)estimated" is perhaps better than "over-(under-)represented"

Scatter plots in Fig.1

In all the plots except (c) the variables on the vertical axis are correlated with the variable of the horizontal axis. This is most obvious for Fig. (a) where hair-HOAPS is used in both abscissa and ordinate. In such cases the binning according to one axis can show biases that are not necessarily real. Whether this is really the case can be easily demonstrated by making the same plot but now with hair-insitu on the horizontal axis. Similarly unrealistic bias may be seen in (b) and (d) because wind and wvpa are derived with from the same satellite channels and therefore correlate with hair-HOAPS. Please discuss.

Page 9, Line 21

Please specify what "even stronger winds" are.

Page 9, Lines 24-26

This paragraph is hard to read. After reading, a number of times times, I think I understand. Is it not better to say: "Our goal is to document the upper bound of the bias and therefore we take the absolute value of the possible systematic error in CE"?

Page 10, line 15 and page 11, line 7

I suggest to replace "Next to" by "In addition to"

Page 11, section 3.5

This section is hard to read. If I understand correctly, it addresses the question: Does it matter for the averages that the satellites sample the ocean at particular times of the day only, given that a diurnal cycle may be present? The authors investigate by looking at buoy data and by comparing averages that cover the full diurnal cycle with samples at satellite overpass times only. Part of the confusion is because it mentions spatial sampling, but I don't think this section covers that? Please simplify for clarity.

page 13, Lines 9-11

I am not sure that it is helpful here to refer to Fig. 1a, because it is showing the combination of $E\_ins(qa\_a)$ and $E\_retr(qa)$, which is different from Fig2_a. The authors point this out but instead of clarifying something it confuses.

Page 13, 23

Suggestion: replace "merely" by "only"

Page 13, Line 24

What is meant by "local minimum in that region for $q\_a$"? $E\_retr(q\_a)$ has a maximum over the warm pool.

Page 13, Line 29

In the sentence "Respective values partly exceed 50 W/m2", what is meant by "respective" and "partly"? Do the authors mean: "In these areas, values are found in excess of 50 W/m2"?

Page 14, Line 33

"direct eddy covariance" is not wind speed.

Page 16, Lines 1-2

This is an interesting example, where it is explained that q_a retrievals may be in error because of dry air advection. However, it is not clear how the systematic error analysis picks up the area of the Agulhas current. The systematic error estimation is entirely driven by U, q_a and SST and wvpa (if I understand correctly).

Section 4.6 and Fig. 4

Here both systematic and random errors are discussed region by region and climatologically versus January/July. Earlier in the paper it was concluded that the random errors were small compared to the systematic errors. However in Fig. 4 the random errors are larger than the systematic errors. Furthermore I would expect that the climatological data (I assume averaged over the entire period) has much more data than the January or July data and therefore much smaller random errors.

Page 18, Line 31

Please replace "outperforms" by "exceeds"

Conclusion:

The material is well worth publishing, but major re-structuring and revision of the paper will improve its quality and accesibility.

---

## Author Comment (AC1) · 1 Dec 2017

**Author's response to the general comments from referee I:**

Thank you for revealing your valuable criticism regarding the manuscript. Below, please find our responses to your specific comments, along with the implemented changes to our manuscript. All page and line numbers as well as figure numbering refer to the *revised* manuscript. Note specifically that the figure numbering has changed during the review process.

**MAIN COMMENTS:**

**i) Main comment from referee:** *However, we are interested in whether the estimated uncertainty is common in (satellite) products or inherent in HOAPS-3.3. If the present results are inherent in HOAPS-3.3, the results are useful for only people to use HOAPS-3.3. However, if the results are common in most satellite products, the value of this article is considerably larger. If possible, we would like to know uncertainties about other products in order to judge whether the estimated uncertainty for HOAPS-3.3 in this study is common or not. I guess it is not so easy for the authors to estimate uncertainties for other products. If so, I would like the authors to investigate the relation between the uncertainties of HOAPS-3.3 obtained by this study and the differences between HOAPS and other products, pointed out by previous paper (Iwasaki et al. (2014)).*

**Author's reponse:** We chose to publish an AMT paper, as our manuscript describes a *technique* for assigning uncertainties to latent heat flux (LHF)-related satellite data. We do not aim at performing an uncertainty assessment of all available data records. Instead, as mentioned in the title, uncertainties are given for HOAPS, which has more than 200 users. We therefore agree that some of our findings cannot be generalized. As is discussed, our displayed uncertainties are in parts related to retrieval uncertainties and sensor noises, which are unique to every data set and satellite instrument, respectively, and are therefore not applicable to other satellite climatologies. We are not aware of any air-sea flux related remotely sensed data set to date that is equipped with instantaneous uncertainty estimates. HOAPS-3.3 therefore leads the way towards a more transparent satellite data analysis, as the user may individually decide how to treat the data, given the available retrieval uncertainties.

More important, we want to highlight the fact that our approach can easily be applied to other satellite data sets, as long as a sufficiently large amount of collocations can be achieved. Choosing a similar in situ data basis and identical collocation criteria compared to our manuscript, random in situ (here: $E_{ins}$) and collocation uncertainties (here: $E_c$) are thought to be comparable to our results, independent of the investigated satellite climatology. As you state, this considerably increases the value of this article.

The uncertainty estimates cannot be set into relation with other satellite climatologies, as no further uncertainty values exist for comparison. However, we agree that the research community would benefit from investigations answering the questions „Do other LHF-related data sets lie within the uncertainty range specified by HOAPS-3.3? If not, how can we explain this discrepancy?". As noted in Sect. 4.7 of the present manuscript, we are currently preparing a follow-up publication regarding this aspect. It will present our findings in a larger perspective and thus increase the importance of our uncertainty analysis.

To increase the value of the present manuscript, we have established links to E-P intercomparisons illustrated by Iwasaki et al. (2014) whenever it fits the context (see „changes" below). One must keep in mind, however, that HOAPS-3 (as is used in Iwasaki et al. (2014)) differs from HOAPS-3.3 used within the present work. Apart from a temporal extension by seven years, this includes changes to the calibration model of SSM/I brigthness temperatures (Fennig et al. (2013)), an updated version of the AVHRR Pathfinder Data Set (SST), and the inclusion of SSMIS data (Fennig et al. (2015)).

Iwasaki et al. (2014) is valuable when it comes to intercomparing various freshwater flux products over the global ice-free oceans. It identifies individual parameter contributions to the overall observed differences and allows for assessing which parameters contribute most to the positive

trend in E. Yet, such an intercomparison does not allow for drawing conclusions regarding the uncertainty of the individual data sets. Observed differences between two data sets could either diminish or amplify when applied to the respective climate data set. In this regard, the present manuscript is very progressive, as it sets a basis for assigning uncertainty measures to climate data records. For example, our uncertainty estimates allow for concluding whether the illustrated differences were to be expected or not. Large differences, coupled to small HOAPS-3.3 uncertainty estimates, would point at retrieval issues related to the data set compared to.

**Changes in the manuscript:** Iwasaki et al. (2014) is cited for the first time in Sect. 1 (P.3, L.27). In the following places of Sect. 4, it is picked up again, where is relates HOAPS to other LHF climatologies: P.17, L.5/25; P.20, L.2; P.21, L.24/26.

Sect. 1 (P.4, L.29) now includes a sentence, which emphasize the fact that the methodology may easily be transferred to other retrievals, which increases the value of our manuscript. This is revisited in Sect. 5 (P.20, L.27) and also implemented in the abstract (P.1, L.17).

**ii) Main comment from referee:** *This article is based on Kinzel et al. (2016). However, the paper is not referred in the present introduction. It is curious. The purpose of this article is not so clear for me. I think the purpose of this study is comprehensive estimation of uncertainty characterization of HOAPS-3.3 latent heat flux (LHF) related parameters in addition to specific humidity examined in Kinzel et al. (2016).*

**Author's reponse:** We agree that the introduction benefits from citing Kinzel et al. (2016), as it introduces the concept of random uncertainty decomposition, which is perfomed within the present study. The approach presented in Kinzel et al. (2016) should be understood as one of several *prerequisites* for our work, as it a) (only) focuses on random uncertainties and b) does not cover wind speed (U), LHF, and evaporation (E). We will provide a citation in an appropriate place und put Kinzel et al. (2016) into a larger context.

**Changes in the manuscript:** Kinzel et al. (2016) is now referenced in Sect. 1 (P.4, L.10/21), where it is also put into a larger perspective.

**iii) Main comment from referee:** *For example, the authors attribute the global minimum during boreal summer 1991 to the Mount Pinatubo eruption. However, we cannot find the minimum in 1991 in other products except HOAPS (Iwasaki et al. (2014, their Fig.6a). Therefore, the minimum may be due to the HOAPS retrieval error related to the Mount Pinatubo eruption.*

**Author's reponse:** Regarding the 1991 minimum related to the Mount Pinatubo eruption: we agree on this. Please refer to the specific comment #20 further below for more details on this. The explanation for the SST feature seen in HOAPS LHF during 1991 was already implemented in the submitted version (see P.18 ,L.25f of revised manuscript).

**iv ) Main comment from referee:** *Also, since all HOAPS parameters are derived from SSM/I and SSMIS microwave radiometers, the sampling errors are expected to be large compared with other products using many kinds of microwave radiometers.*

**Author's reponse:** We agree that differences in sampling between different instruments exist, which may cause sampling biases. However, it should be kept in mind that the manuscript demonstrates an application of the introduced methodologies and does not focus on an assessment or intercomparison of sampling uncertainties. The SSM/I and SSMIS sampling uncertainties are accounted for, which play a marginal role on climatological time scales. This is mirrored in the small magnitudes of monthly mean $E_{smp}$ in Table 2 of the present manuscript.

**v) Main comment from referee:** *Although the second paragraph in the section 5 introduces HOAPS 4.0, I feel the paragraph is not necessary in this section.*

**Author's reponse:** We believe it is important to note that the newest version of HOAPS, that is HOAPS 4.0 (released in October 2017), includes an update of the uncertainty estimates. Apart from this, we outline new features and improvements with respect to HOAPS-3.3 in two sentences. We

agree that it is somewhat out of place in the submitted manuscript. This short paragraph is therefore placed towards the end of Sect. 5.
**Changes in the manuscript:** The short paragraph related to HOAPS 4.0 has been moved to the end of Sect. 5 (P.22, L.6ff).

**vi) Main comment from referee:** *Moreover, the authors discuss about precipitation in this section, but I think this issue may exceed the scope of this study because they do not carry out uncertainty estimates of HOAPS precipitation here.*
**Author's reponse:** We generally agree with this comment and will therefore remove parts of the provided literature review on issues with satellite precipitation (P) estimates. However, we want to continue emphasizing the importance of quantifying P uncertainties, because it ultimately allows for assessing uncertainties in freshwater budgets (E-P). In this context, the mentioned study by Burdanowitz et al. (2016) is valuable, as it lays the basis for this purpose.
**Changes in the manuscript:** The paragraph related to uncertainties in P has been shortened (P.21, L.30ff).
* * *
**SPECIFIC COMMENTS:**

**1 )Comment from referee:** P.1, L.1: "of LHF" → "of in situ LHF"
**Author's reponse:** We agree that 'in situ' should be added in this context
**Changes in the manuscript:** 'in situ' has been added to the revised manuscript (P.1, L.1).

**2) Comment from referee:** *P.3, L.21-27: In this paragraph, we need clear description about characteristics related to uncertainties, of HOAPS LHF product compared with other products obtained by numerous intercomparison studies*
**Author's reponse:** The mentioned paragraph serves to merely introduce the HOAPS climatology. Apart from listing included parameters, the brief literature review on HOAPS intends to demonstrate its usefulness in climate research and highlight its performance in context of intercomparison studies. For further information, the reader is referred to the quoted references. We believe that a thorough description related to uncertainty characteristics exceeds the scope of this introductory paragraph. However, we agree that highlighting some distinct differences among the data sets (without a focus on uncertainty estimates) would improve our introduction.
**Changes in the manuscript:** A paragraph has been added to the revised manuscript (P.3, L.25ff), which points at substantial differences between LHF data sets (including HOAPS) on a local scale. A second paragraph deals with performed uncertainty characterizations related to LHF (P.4, L.4ff). It shows what has been done to date and points at the shortcoming that, apart from NOCS v2.0, no uncertainty estimates are available to the users.

**3) Comment from referee:** *P.5,L.16-21: Large El Nino and La Nina occurred in 1997-1998. Therefore, 1997-1998 is a special period. Why did the authors use the data in this period?*
**Author's reponse:** We agree that 1997-1998 were "special" years, in a climatological sense. We argue that for training purposes, it is not essential whether the contributing data was obtained during climatologically anomalous years or not. What counts is that a) the network is trained with match ups, which are physically connected and b) the whole possible range of atmospheric conditions (i.e., in this case wind speeds) is covered by a representative amount of data. It that sense, match-ups from 1997-1998 are beneficial, as they guarantee a full coverage of all conditions. Thus, potential extremes are covered in our training data base.

**4) Comment from referee:** *P.5,L.33-34: The assumption of a constant relative humidity of 80 % and air-sea temperature difference of 1 K is considerably artificial. To what extent does have the assumption impact on estimation of uncertainty?*

**Author's reponse:** Thank you for bringing this up. We did not investigate the uncertainty introduced by these two widely used assumptions, as it may be neglected for two reasons. First, air temperature only has a secondary effect on LHF (in contrast to SHF) through the stability of the atmospheric column. At the same time, the assumption of 1 K temperature difference with respect to SST is a good approximation for vast regions over the global oceans. However, we agree that over upwelling regimes, which are very confined compared to the global oceanic area, this approximation is violated. Compare conclusion section of Wells and King-Hele (1990). Second, our uncertainty estimation procedure described in Sect. 3 is exclusively based on high-quality match-ups of HOAPS and in situ measurements. The data density of both ship and buoy records is comparably low in the upwelling regimes, which further reduces the impact of our two assumptions. Due to the comparatively small amount of reference data, we presumably underestimate resulting uncertainties in these regions. Using for example ancillary reanalysis-based data would violate our ambition to create a completely remotely-sensed data record, which is a key feature of HOAPS.

**5) Comment from referee:** *P.6,L.25: (2003) ---(2013)*
**Author's reponse:** We agree that Bentamy et al. (2013) is worth citing here.
**Changes in the manuscript:** A citation of Bentamy et al. (2013) has been added to revised manuscript (P.7, L.3).

**6) Comment from referee:** *P.8,L.15: In what ways are these features similar?*
**Author's reponse:** The term 'similarity' refers to the similarity of the bias distributions as a function of the x-axis parameters. That is, lowest SST (i.e., high latitudinal SST) are underestimated in HOAPS (likewise, $q_a$ is underestimated for (high latitudinal) $q_a$ below 5 g kg$^{-1}$). The HOAPS underestimation also accounts for subtropical SST in the range of 25°-29°C (likewise, $q_a$ is underesimated for $q_a$ between 15-19 g kg$^{-1}$). By contrast, HOAPS SST are slightly overestimated for SST ranging between approximately 15°-24°C and the inner tropics (30°C). Likewise, $q_a$ is overestimated for $q_a$ between 7-12 g kg$^{-1}$ and for inner tropical 20 g kg$^{-1}$).
**Changes in the manuscript:** The wording has been modified in the revised manuscript (P.9, L.14f).

**7) Comment from referee:** *P.8,L.22: " off the Arabian Peninsula". We cannot recognize the data off the Arabian Peninsula in Fig. 1. We need the distribution of average $q_a$ for this.*
**Author's reponse:** Indeed, thank you for pointing this out. This paragraph is meant to exemplarily present the benefit of multi dimensionality, whereas the illustration of $q_a$ patterns referred to is not the primary focus. We therefore omitted an additional map showing the distribution of $q_a$ and U over the Arabian Sea. However, we have included a global map showing the average difference between HOAPS and in situ $q_a$
**Changes in the manuscript:** 'not shown' has been added twice to the revised manscript (P.9, L.24f).

**8) Comment from referee:** *P.9 L.11: Is the bin width equal or not? How did you determine the bin width?*
**Author's reponse:** The bin width is not equidistant. It is rather determined by fixed percentiles of data, where 5% of all contributing match-ups are assigned to a single bin. In consequence, 20 bins result, which are narrow for large data densities and become wide close to the tails of the distribution. This is also picked up it the caption of Fig. 2.
**Changes in the manuscript:** A note on the bin configuration has been added to the revised manscript (P.9, L.3ff) and is again picked up in context of Sect. 3.2 (P.10, L.16) and the caption of Fig. 2.

**9) Comment from referee:** *P.9, L.17: Why did you choose the different data period between ($dq_a$, dU) and ($dq_s$)?*

**Author's reponse:** For $dq_a$ and $dU$, the vast amount of in situ data justified the restriction to collocations between 2000 and 2008. For $d_{qs}$, the time period from 2002-2005 was left out, as corresponding local equatorial overpasses of the operating NOAA-17 were disadvantageous for our double collocation analysis. Recall that only *night-time* SST were collocated to in situ measurements to avoid the warm layer effect (see Sect. 2.2 of this manuscript). Fulfilling the requirement of local night time, the overpass times of NOAA-17 were inappropriate for gathering a large number of collocations. Instead, collocation during 2006-2008 were used. Additionally, the period 1998-2001 was taken as reference to allow for a sufficiently large collocation data basis. In consequence, $dq_s$ match ups are based on collocations from 7 years only. This does not pose a problem, as in situ SST measurements were available more frequently compared to U and $q_a$.

**10) Comment from referee:** *P. 10, L.4: The average of daily coefficients is applied for estimation of instantaneous LHF uncertainties here. Why are not instantaneous values but daily values applied? Also, is the difference between daily and instantaneous coefficients small or large?*
**Author's reponse:** Thank you for bringing this up. We had similar thoughts regarding the representativity of daily versus instantaneous correlations. Deriving instantaneous correlation coefficients, however, has a key disadvantage. Most of the global ocean is scanned only 1-2 times per day by a single SSM/I or SSMIS instrument, some regions over the subtropics not at all. This implies that the amount of instantaneously derived geophysical parameters is locally very limited. Resulting correlation coefficients would therefore not be representative. We therefore decided to apply global averaged coefficients, which are remarkably stable throughout the year on a day-to-day basis (not shown). Due to this decision we are not capable of comparing our coefficients to instantaneous correlation coefficients. We are aware that differences may occur.
However, we furthermore investigated, how much the sum of all correlation terms in Eq. (2) contributes to instantaneous $\sigma_{LHF,sys}$. On average, omitting these correlation terms modifies the resulting instantaneous $\sigma_{LHF,sys}$ by merely $0.5 \pm 5$ W m$^{-2}$. Thus, even if global mean correlation coefficients were not always the most accurate choice, they do not represent a key contribution to resulting LHF uncertainty estimates.
**Changes in the manuscript:** The two reasons for why we apply the average of daily mean global correlation coefficients have been included into the revised manuscript (P.11, L.31ff).

**11) Comment from referee:** *P.10,L.8: Could you explain about the definition of "gridded uncertainty products"?*
**Author's reponse:** Sorry for not being precise here. By „gridded products", we general mean satellite data that has been spatially and temporally averaged and that is available for fixed grid cells (dx,dy) and time periods (dt), like 'HOAPS-C' and 'HOAPS-G'. This stands in contrast to instantaneous, level-2 data (points in time and space, like 'HOAPS-S'), which form the basis of our uncertainty analysis. To avoid confusion, we will not mention this in the revised manuscript and rather rephrase this sentence.
**Changes in the manuscript:** The wording has been modified in the revised manuscript (P.12, L.8f)

**12) Comment from referee:** *P.10,L.17: What is a true value for $E_c$?*
**Author's reponse:** We do not understand the question. Please see Table 1 of our manuscript for magnitudes of HOAPS LHF-related $E_c$ resulting from the random uncertainty decomposition.

**13) Comment from referee:** *P.11,L.19-22: Here, all daily sampling uncertainties are derived as a function of the number. However, sampling error for a daily-mean value depends on not only the number but also observation times.*
**Author's reponse:** This is absolutely correct. Assuming a specific number of daily overpasses was a prerequisite for showing the sampling uncertainties as a function of operating satellites (Table 2 of our manuscript). P.11, L.18 of the submitted manuscript indicates that the daily sampling uncertainties are estimated using "simultated satellite records", which are derived using the two

buoy records closest in time to local satellite overpasses. The assumption of having two overpasses per day is reasonable, as this applies to vast regions of the global oceans. We assume that sampling uncertainties are inverse proportional to the amount of daily overpasses, but do not investigate this dependency further. As the number of daily overpasses increases with an increasing number of satellites, we rather resolve the resulting sampling uncertainties as a function of orbiting platforms. This is in line with conclusions by Tomita and Kubota (2011), who found that multi-satellite simulations for e.g. $q_a$ considerably reduced the sampling uncertainty, compared to single satellite simulations.

**Changes in the manuscript:** The wording has been modified in the revised manuscript (P.12, L.18f) to point out that our estimates are based on the assumption of having two overpasses per day.

**14) Comment from referee:** *P.12,L.1-3: We find several geographical words such as " Arctic", " polar" and " inner tropics". However, it is difficult for us to obtain the relation between the ranges of the random satellite retrieval uncertainty and the geographical location from Fig. 1 and Table 1. Also are the values shown in this paragraph consistent with those in Table 1? For example, " 0.3 and 1.8g kg$^{-1}$" is " 0.7 and 1.8g kg$^{-1}$" in line 1?*

**Author's reponse:** Thank you for pointing this out. We agree that this is confusing and will clarify this in the revised manuscript, as Table 1 does not show distributions of the random retrieval uncertainty as a function latitude and longitude. Regarding the consistency of values shown in Fig. 2 and Table 1: Note that directly comparing results of Table 1 to Fig. 2 (and expecting equality) is not correct. Fig. 2 shows bin-wise biases and their spread in *one-dimensional* space. The values in Table 1, however, result from the *multi-dimensional* bias analysis, multiple triple collocation (MTC) analysis, and subsequent random uncertainty decomposition. This implies that random retrieval uncertainties of $q_a$ presented in Table 1 are compatible with the global distributions shown in Fig. 3a. Regarding 0.3 g kg$^{-1}$ vs. 0.7 g kg$^{-1}$: we apologize for this mistake, '0.3' is a typo and has been corrected to 0.7 g/g kg$^{-1}$ in the revised manuscript.

**Changes in the manuscript:** The geographical terms have been removed and have been replaced with $q_a$ magnitudes (P.13, L.10ff). The typo has been corrected (P.13,L.10). The captions of Table 1 and Fig. 3 have been modified to point at the similarity of both representations (i.e., showing $E_{retr,ran}$).

**15) Comment from referee:** *P.12, L.1-28: Accuracy of in situ data is considerably different depending on used sensors. For example, the accuracy of wind speeds is 1.0m/s or 10% for usual NDBC buoys, while that is 0.3 m/s for TOA buoys. Are these differences between them negligible for the present analysis?*

**Author's reponse:** Thank you for providing this differentiation regarding accuracies of buoy measurements. Sect. 4.1 deals with the random uncertainty component (that is, precision) and does not target accuracies. However, we generally agree that different instruments are associated with a variety of (random) measurement uncertainties. Sect. 4.1 (and thus Table 1) results from a random uncertainty decomposition procedure (compare Kinzel et al. (2016)), which crucially depends on the amount of contributing triple collocations and thus in situ measurements. Our collocation data basis is very large, including a variety of exclusively high-quality in situ measurements. The results of the decomposition itself should be interpreted in a way, such that *average* random insitu measurement errors can be separated from *average* random retrieval and collocation uncertainties, depending on the magnitude of $q_a$, U, and $q_s$. See for example the orange, red, and black squares as a funtion of $q_a$ in Fig. 2 of Kinzel et al. (2016) for an illustration of this decomposition. Each of these orange squares can be understood as a *bin-averaged* random in situ uncertainty contribution. Thousands of in situ data records contribute to each of these squares/bins. One needs to therefore consider our random in situ uncertainties as an *average* over all in situ data sources for a specific parameter regime, i.e., bin. Therefore, individual in situ accuracies do not receive much weight.

**Changes in the manuscript:** Sect. 3.3 has been extended by two sentences (P.11, L.15ff), which emphasize that the random uncertainty magnitudes illustrated in Table 1 are derived bin-wise and

result from thousands of triple collocated match ups (and thus in situ records).

**16) Comment from referee:** *P.14, L.13: Could you tell me the definition of the climatological total uncertainties ($E_{clim}$)? Are the climatological total uncertainties ($E_{clim}$) different from the systematic uncertainty?*
**Author's reponse:** Sorry for not being precise enough here. For each grid box of Fig. 3, we define the climatological uncertainty ($E_{clim}$) as the mean root mean squared sum of $E_{sys}$, $E_{retr,ran}$, and $E_{smp}$ (1988-2012). As $E_{retr,ran}$ scales with 1/N, with N being the amount of observations per grid box, it becomes virtually zero for the temporal averages shown in Fig. 4 of our manuscript. Likewise, monthly mean $E_{smp}$ are small (see Table 2 of our manuscript). Thus, on climatological timescales, $E_{clim}$ and $E_{sys}$ do not differ. We will emphasize the definition of $E_{clim}$ more clearly in the revised manuscript.
**Changes in the manuscript:** A sentence has been added to Sect. 4.3 (P.15,L.12ff), which explains our definition of $E_{clim}$. It has also been added to the caption of Fig. 4.

**17) Comment from referee:** *P.16,L.28: What is the meaning of "isolated time periods"?*
*Author's reponse: We apologize for not being precise here. This was to state that during individual months ('isolated time periods'), the global mean uncertainty (one value) deviates from the respective average of 1988-2012.*
**Changes in the manuscript:** The wording has been modified in the revised manuscript (P.18, L.9f).

**18) Comment from referee I:** *P.17, L.3-19: $E_{clim}$ is considered to be only one value from the meaning of a climatological value. Is it right? If so, I cannot understand the meaning of " respective $E_{clim}$ over the Pacific upwelling regimes reaches 25 W m$^{-2}$ specifically during boreal spring 1998" found in line 6-7.*
**Author's reponse:** $E_{clim}$ is defined separately for each grid box (see comment #16 above on this), which is why we are explicitly able to e.g. point at climatological uncertainties over the Pacific upwelling regime.
**Changes in the manuscript:** See comment #16 above.

**19) Comment from referee:** *P.17, L.28: "climatological regional wind speeds range between 4.5.-11 m s$^{-1}$ (fig.4b). As for $q_a$"* --> *"climatological regional uncertainties in wind speeds range between 4.5.-11 m s$^{-1}$(fig.4b). As for U"*
**Author's reponse:** We are not sure whether we understand this comment correctly. As formulated, the range of 4.5-11 m s$^{-1}$ consideres the regional wind speed *itself*, not its related uncertainties. Fig. 4b shows regional and global mean HOAPS U, along with systematic and random retrieval uncertainties. The individual medians range between 4.5-11 m s$^{-1}$. Seasonality is most pronounced over the Indian monsoon region, WBC, and the North Atlantic (see JJA and DJM in Fig. 4b for this). Similar conclusions can be drawn for $q_a$ (Fig. 4a), regarding maxima in seasonality for those three regions.

**20) Comment from referee:** *P.18, L.10: The global minimum during boreal summer 1991 is linked to the Mount Pinatubo eruptions. However, the remarkable minimum can be found in only HOAPS product and cannot be found in other products as shown in Fig. 6(a) of Iwasaki et al. (2014). Therefore, the minimum would be related to retrieval model uncertainty. The present analysis can investigate this issue and present its effectiveness by the investigation.*
**Author's reponse:** Thank you very much for pointing at the valuable study by Iwasaki et al. (2014), which we missed to cite so far. Indeed, the global minimum is linked to the Mount Pinatubo eruption and is not observed in the remaining satellite and reanalysis products. Similar to our work, the authors point at the cause of this low bias, which is attributed to AVHRR aerosol issues. In consequence, this created low-biased SST (i.e., low-biased $q_s$), which in turn resulted in

unrealistically low near-surface humidity gradients and thus low-biased E. This has already been picked up in e.g. Andersson et al. (2010) and is therefore a known issue related to the retrieval model. The recently released HOAPS 4.0 climatology (Andersson et al., 2017) does not include this feature anymore, as the SST reference has changed to the NOAA 0.25° daily Optimum Interpolation Sea Surface Temperature (OISST, Reynolds et al. (2007)), which corrects for this effect (see Reynolds, 1993). We are not aware of further systematic retrieval issues and the overall good performance of HOAPS in relation to other satellite and reanalysis data sets is mirrored in e.g. Iwasaki et al. (2014) (e.g. their Fig. 3). Regarding the classification of the low-biased LHF during 1991 (see Fig. 5 of our manuscript) with respect to the given uncertainty ranges: the low-biased LHF lies within the average HOAPS LHF retrieval uncertainty range (gray shading) between 1988-1998.

**Changes in the manuscript:** The explanation for the SST feature seen in HOAPS LHF during 1991 was already implemented in the submitted manuscript (P.18, L.25ff of revised manuscript). Furthermore, Iwasaki et al. (2014) has been included to the reference list and is cited where appropriate throughout the revised manuscript (see general comment #1 at the top of this document for more details).

**21) Comment from referee:** *P.18,L.15: As mentioned before, could you please explain about definition of climatological uncertainty? I cannot catch the meaning of " the 12-month running mean climatological uncertainty". Is a climatological uncertainty defined each month?*

**Author's reponse:** See comment #16 and #18 regarding the definition of $E_{clim}$. From these grid point wise $E_{clim}$, a global mean climatological uncertainty is derived for each month. This implies that twelve values result for each year. For smoothing purposes, an annual (that is , 12-month) running mean is performed over these 25x12 = 300 global monthly mean values.

**Changes in the manuscript:** The wording which describes Fig. 6 has been modified in the revised manuscript (P.18, L.29ff). Keeping the definition of $E_{clim}$ in mind (see comments #16 and #18), it becomes clear that a global mean value of $E_{clim}$ can be calculated for each month, to which running means can be performed. Furthermore, the caption of Fig. 6 has been slightly adjusted.

**22) Comment from referee:** *P.18, L.21-P.19,L. 5: In this paragraph, the results by many previous studies are introduced. However, the relation between the results and what Fig.5 shows is not so clear. I wonder this paragraph is necessary.*

**Author's reponse:** We agree that the focus of our manuscript lies on the uncertainty characterization, rather than on the positive trend seen in LHF.

**Changes in the manuscript:** The respective paragraph has been shortened (P.20, L.1-6).

**23) Comment from referee:** *Fig 2. (c) and Fig. 3. (c) It is difficult to know the distribution pattern in these figures. How about the change of a color bar?*

**Author's reponse:** These colorbars were chosen in order to be identical to the colorbars of Fig. 3a and 4a, respectively. Doing this, one can directly see the comparatively small uncertainty contributions of $q_s$ in relation to $q_a$. Specifically regarding Fig. 3c, the distribution may not always be distinct. However, the most important feature in Fig. 3c, that is the maximum over the Indo-Pacific warm pool region, is well resolved. Pattern descriptions are additionally given for Fig. 3c (P.14, L.30-34) and Fig. 4c (P.16, L.20-25)

**Changes in the manuscript:** A comment regarding the same color bar range of $q_a$ and $q_s$ has been included in the figure captions of Fig. 3 and 4.

**Cited studies:**

**Andersson**, **A.**, Graw, K., Schröder, M., Fennig, K., Liman, J., Bakan, S., Hollmann, R., Klepp, C.: Hamburg Ocean Atmosphere Parameters and Fluxes from Satellite Data - HOAPS 4.0, Satellite Application Facility on Climate Monitoring, doi: 10.5676/EUM_SAF_CM/HOAPS/V002, 2017

**Fennig, K.**, Andersson, A., and Schröder, M.: Fundamental climate data record of SSM/I brightness temperatures, doi:10.5676/EUM_SAF_CM/FCDR_SSMI/V001, 2013.

**Fennig, K.,** Andersson, A., and Schröder, M.: Fundamenal Climate Data Record of SSM/I / SSMIS Brightness Temperatures, CM SAF, doi: 10.5676/EUM_SAF_CM/FCDR_MWI/V002, 2015

**Iwasaki, S.,** Kubota, M., and Watabe, T.: Assessment of various global freshwater flux products for the global ice-free oceans, Remote Sensing of Environment, 140, 549-561, doi: 10.1016/j.rse.2013.09.026, 2014.

**Reynolds**, **R. W.**: Impact of Mount Pinatubo Aerosols on Satellite-derived Sea Surface Temperatures, Journal of Climate, 6, 768-774, doi: 10.1175/1520 0442(1993)006<0768:IOMPAO>2.0.CO;2, 1993.

**Reynolds, R. W.,** Smith, T. M., Liu, C., Chelton, D. B., Casey, K., and Schlax, M. G.: Daily High-Resolution-Blended Analyses for Sea Surface Temperature, J. Climate, 20, 5473–5496, doi: 10.1175/2007JCLI1824.1, 2007.

**Tomita, H.** and Kubota, M. K.: Sampling error of daily mean surface wind speed and air specific humidity due to sun-snychronous satellite sampling and its reduction by multi-satellite sampling, International Journal of Remote Sensing, 32, 3389–3404, doi: 10.1080/01431161003749428, 2011.

**Wells, N.** And King-Hele, S.: Parameterization of tropical ocean heat flux, Quart. J. Roy. Meteor. Soc., 116, 1213–1224, doi: 10.1002/qj.49711649511, 1990

---

## Author Comment (AC2) · 1 Dec 2017

**Author's response to the general comments from referee II:**

Thank you for revealing your valuable criticism regarding the manuscript. Below, please find our responses to your specific comments, along with the implemented changes to our manuscript. All page and line numbers as well as figure numbering refer to the *revised* manuscript. Note specifically that the figure numbering has changed during the review process.

**SPECIFIC COMMENTS:**

**1 ) Comment from referee:** *Page 5, Line 10, "... the pixel-level HOAPS-3.3 data in sensor resolution is used...". What are the spatial and temporal resolutions of the pixel-level HOAPS-3.3 data? Which nine sensors are used in the pixel-level HOAPS-3.3 climatology?*
**Author's reponse:** The spatial resolution of both data sources is channel-dependent. For SSM/I data (DMSP F08-F15), it varies from 69 km by 43 km (19 GHz channel) to 37 km by 28 km (37 GHz channel). Sampling frequencies take on a value of 25 km, corresponding to scan lines every few seconds. Regarding SSMIS (DMSP F16-F18): The spatial resolution varies from 74 km by 47 km (19 GHz channel) to 41 km by 31 km (37 GHz channel). As for SSM/I, sampling frequencies are given by 25 km. Overall, 9 different DMSP sensors contribute to HOAPS-3.3: F8, F10-F11, and F13-F18.
**Changes in the manuscript:** The DMSP satellite platforms have been included into the revised manuscript (P.5, L.10f). Furthermore, the spatial resolution has been implemented (P.5, L.11ff).

**2) Comment from referee:** *Page 5, Line 15: what is the temporal resolution of $q_a$ retrievals? And at what height?*
**Author's reponse:** Unfortunately, no information is provided by Bentamy et al. (2003) as to the sensor heights of the (in situ) $q_a$ retrievals. What is known is that their updated regression coefficients are derived using 1000 collocations between globally distributed ship data and validated DMSP satellite data (F10-F14) during 1996-1997. As the retrieval is based on these match ups, we believe that the expression of "temporal resolution" is somewhat misleading. The globally distributed match ups do not have a temporal resolution and are rather point measurements in time and space.

**3) Comment from referee:** *Page 5, Line 32: which surface pressure data are used in computing LHF?*
**Author's reponse:** The COARE-3.0 algorithm assumes a standard sea level pressure (SLP) of 1013.25 hPa when iteratively calculating LHF, which is also used for deriving HOAPS LHF. Brodeau et al. (2017) investigated the effects of this SLP approximation in bulk parameterizations of tubulent air-sea fluxes, amongst others. The authors conclude that errors of such an approximation remain well below discrepancies related to the computation of the transfer coefficients themselves. Their sensitivity experiments show that $q_s$- and $\rho$-induced errors range between merely ±5% (given an SLP range from 950 hPa to 1040 hPa) with an opposite and therefore potentially compensating effect on LHF. Apart from this, both SSM/I and SSMIS are not capable of deriving SLP. Making use of auxiliary (e.g. reanalysis) data to implement SLP would violate HOAPS' unique feature of relying completely on satellite input.
**Changes in the manuscript:** A note has been added to the revised manuscript (P.6, L.9) that a constant SLP is presumed.

**4) Comment from referee:** *Page 5, Line 33: "...surface air temperature, which is estimated by assuming a constant relative humidity of 80 % (Liu et al., 1994) and air-sea temperature difference of 1K". How accurate is this assumption? During winter cold air outbreaks over the western boundary current regions, the air-sea temperature differences can exceed 10 K. In this case, the*

*assumption will lead to a bias in air temperature. How is surface air temperature compared to the in situ dataset?*

**Author's reponse:** Thank you for bringing this up. We did not investigate the uncertainty introduced by these two widely used assumptions, as it may be neglected for two reasons (for our purposes).

First, air temperature only has a secondary effect on LHF (in contrast to SHF) through the stability of the atmospheric column. The assumption of 1 K temperature difference with respect to SST is a good approximation for vast regions over the global oceans. However, we agree that during cold air outbreaks over the WBCs or in upwelling regimes, which are very confined compared to the global oceanic area, this approximation is violated. Compare conclusion section of Wells and King-Hele (1990).

Second, our uncertainty estimation procedure described in Sect. 3 is exclusively based on high-quality match-ups of HOAPS and in situ measurements. The data density of both ship and buoy records is comparably low in the regions addressed above, which further reduces the impact of our two assumptions. Due to the comparatively small amount of reference data, we presumably underestimate resulting uncertainties in these regions. Using for example ancillary reanalysis-based data would violate our ambition to create a completely removely-sensed data record, which is a key feature of HOAPS.

No SSM/I or SSMIS retrievals exist that are capable of accurately retrieving oceanic surface air temperature (SAT) from space. This implies that SAT is not available as an official HOAPS product and has thus not been compared to the in situ reference. Future efforts will take on this challenge.

**5) Comment from referee:** *Page 6, Line 1: Provide a map showing the spatial distribution of in situ (ship and buoy) reference data density over the global domain.*

**Author's reponse:** We agree that providing such a map is useful to the reader. We therefore implemented a map showing the spatial distribution of match ups (ship/buoy vs. satellite) over the global oceans, exemplarily for $q_a$. It shows all collocated match ups between 2001-2008 that contribute to Fig. 2 ($\approx$ 13.8 million match ups per subplot in total). Match ups for U and $q_s$ occur even more frequently, but are not shown in the revised manuscript.

**Changes in the manuscript:** A map showing the distribution of $q_a$ collocations between 2001-2008 has been implemented into the revised manuscript (Fig 1, left panel). It is briefly described in terms of density distributions (P.8, L.29ff).

**6) Comment from referee:** *Page 6, Lines 4-5: Does the reference dataset include the 1996-97 period that is used in training $q_a$ algorithm?*

**Author's reponse:** We are not able to answer this question, as Bentamy et al. (2003) does not provide any information as to which ship records were used to train their $q_a$ retrieval. Yet, the multi-dimensional bias analyses are restricted to match ups between 1998 and 2008 (depending on the parameter, see P.10, L.25f). This implies that no temporal overlap between the reference data archive and the ship records used for training purposes exists.

**7) Comment from referee:** *Page 7, Lines 28-29: The "instantaneous and climatological uncertainties" are not explained. How are they related to systematic, random, and sampling uncertainties?*

**Author's reponse:** Sorry for not being precise enough here; we agree that this needs clarification. "Instantaneous" uncertainties are pixel-level uncertainties. These uncertainties can either be systematic (compare Fig. 4 over revised manuscript) or random (see Fig. 3 of revised manuscript). On an instantaneous basis, sampling uncertainties do not exist.

By contrast, we define "climatological" uncertainties as *total* uncertainties averaged over the time period 1988-2012 (as illustrated in Figs. 4 and 5 of revised manuscript). That is, $E_{clim}$ is formally the mean root mean squared sum of $E_{sys}$, $E_{retr,ran}$, and $E_{smp}$ averaged over 1988-2012. As $E_{retr,ran}$ scales with 1/N, with N being the amount of observations per grid box (see Eq. 3), it becomes virtually

zero when averaging over long time periods. Likewise, monthly mean $E_{smp}$, which applies even more so to multi-annual averages. On climatological time scales, $E_{clim}$ and $E_{sys}$ therefore hardly differ. This is why Fig. 4 of the revised manuscript can be treated as both „systematic" and „climatological" uncertainty.

**Changes in the manuscript:** The explanation of the methodology has been extended (P.8, L.12ff). This includes a link from instantaneous and climatological uncertainties to systematic, random, and sampling uncertainties. A mathematical description of $E_{clim}$ is furthermore provided (P.15, L.12f).

**8) Comment from referee:** *Page 8, Line 10: Definition of water vapour path?*
**Author's reponse:** The water vapour path ("wvpa") refers to the vertically integrated water vapour and is therefore a measure of humidity contents in the atmospheric column. It is thus suitable to use as an indicator of the ambient atmospheric conditions. For more information regarding the HOAPS-3.3 wvpa retrieval, please refer to Schlüssel and Emery (1990).
**Changes in the manuscript:** The term „"water vapour path" has been replaced by "vertically integrated water vapour" (P.9, L.2).

**9) Comment from referee:** *Page 8, Lines 11-14: It seems that HOAPS $q_a$ is wet biased in the tropical wet zone and dry biased in the subtropical dry zone. The bias pattern seems to be similar to GSSFT v3 $q_a$ product (Prytherch et al. 2014, Int. J. Climatol.; Jin et al. 2015, J. Atmos. Ocean. Technol.).*
**Author's reponse:** Thank you for pointing this out. Indeed, Figure 4c in Prytherch et al. (2014) shows a strong resemblance between HOAPS-3.2 and GSSTF3. Both data records are based on the same algorithm and follow an inter-satellite calibration procedure. The minor differences in the tropics are thought to be related to either different quality control standards or differing Earth incidence angles. Given the close resemblance of GSSTF3 and HOAPS-3.2 shown in Prytherch et al. (2014), the difference pattern (GSSTF minus buoys and OAFlux) shown in Jin et al. (2015) was to be expected. The distribution is closely related to the $q_a$-dependent bias pattern shown in our manuscript (Fig. 2a).
**Changes in the manuscript:** Prytherch et al. (2014) is cited in this context (P. 9, L.9f).

**10) Comment from referee:** *Page 8, Lines 21-22: Indeed, the 1-D bias analysis is not sufficient. Please provide a figure showing the global pattern of the mean differences between HOAPS and the reference data. Need to discuss the uncertainty pattern in terms of humidity regimes.*
**Author's reponse:** Thank you for your suggestion. Originally we thought the reader would be distracted by such a difference map, as we would like to emphasize the importance of considering *multiple* atmospheric state parameters, i.e., the multi-dimensional bias analysis. However, we agree that the manuscript improves when including such a difference map (HOAPS minus in situ $q_a$).
**Changes in the manuscript:** The difference map has been included into the revised manuscript (Fig 1, right panel). It is briefly described in Section 3.1 (P.9, L.10f,L.25f), where a connection to Fig. 2a (of revised manuscript) is established.

**11) Comment from referee:** *Page 9, Line 24: "Recall that the aim is to characterize uncertainty and not bias patterns". The sentence is confusing. Bias is one kind of uncertainties.*
**Author's reponse:** We disagree with this statement. According to the International Vocabulary of Metrology (VIM, 2012), the (measurement) uncertainty is a *non-negative* parameter characterizing the *dispersion* of the quantity values being attributed to a measurand, based on the information used (VIM, 2.26). By contrast, a (measurement) bias (VIM, 2.18), which corresponds to an estimate of a systematic measurement error (VIM, 2.17), may be either positive or negative and, if known, can be corrected for. Keeping these two definitions in mind, a bias, which is a signed value, is strictly speaking not a kind of uncertainty. In order to turn the bias into an uncertainty estimate, we use the absolute systematic difference as an upper boundary of the (more simple) bias distribution.
**Changes in the manuscript:** The wording in the revised manuscript has been modified and moved

further up in Sect. 3.2 (P.10, L.17-22).

**12) Comment from referee:** *Page 10, Eqs (2)-(3): Which figures are produced from Eqs.(2)-(3)?*
**Author's reponse:** Figs. 3-6 are based on Eqs. 2 and 3. Details are provided in the following. Whereas Eq. 3 merely expresses that the total instantaneous LHF uncertainty consists of a systematic and a random component, Eq. 2 forms the basis of LHF pixel-level uncertainties using uncertainty propagation. That is, applying Eq. 2 equips each LHF pixel with a *total*, that is systematic plus random uncertainty contribution. In consequence, Figure 4d directly results from Eq. 2, that is the systematic uncertainty contribution (the random component convergences to zero, due to averaging over long time period). Likewise, the systematic uncertainty contributions by U, $q_s$, and $q_a$, which contribute to Eq. 2, are illustrated in Figs. 4a-c.
Note that the random uncertainty measures resulting from Eq. 2 still incorporates random uncertainty contributions of the collocated in situ data ($E_{ins}$) as well as the collocation procedure itself ($E_c$). Each random uncertainty contribution resulting from Eq. 2 needs to therefore be corrected to isolate the random *retrieval* uncertainty. This random retrieval uncertainty is what we would like to characterize in the HOAPS climatology. The random LHF uncertainty resulting from Eq. 2 is therefore corrected pixelwise, using the results of the random uncertainty decomposition (see Sect. 3.4 and e.g. Figure 2 in Kinzel et al. (2016) for $q_a$). The average field of these instantaneous, corrected random retrieval uncertainties is shown in Fig. 3d. Respective random retrieval uncertainty components contributed by U, $q_s$, and $q_a$, are shown in Figs. 3a-c, respectively. As noted in the manuscript, Fig. 3 shows the *instantaneous* point of view, that is N=1.
Likewise, Fig. 5 shows both systematic (rectangles) and instantaneous random retrieval (bars) uncertainties. It therefore shows the maximum uncertainty one can expect for a single pixel for different geographical regimes. Figure 5 is therefore based on both Eqs. 2 and 3. The same accounts for Fig. 6. The technical aspects are described in Sect. 3.4-3.5 .

**13) Comment from referee:** *Page 10, Line 10: Why only random satellite retrieval component, not the total random uncertainty, is computed?*
**Author's reponse:** The purpose of our uncertainty characterization is to assign systematic, random, and sampling uncertainties to all *satellite-related* LHF parameters. This approach is unique and important, as simply assigning total random uncertainties does not allow the user to understand to what extent they are associated with the retrieval itself or other uncertainty sources. This implies that contributions by collocation ($E_C$) and in-situ data ($E_{ins}$) need to be corrected for (i.e., removed) by applying the random uncertainty decomposition (Sect. 3.3). What remains is the random retrieval uncertainty, which consists of both random model uncertainty ($E_M$) and sensor noise ($E_N$) (see Kinzel et al. (2016), their Eq. 5).
Immler et al. (2010) formulate an implication of such an approach for consistencies like this: „Roughly speaking, consistency is achieved when the independent measurements agree within their individual uncertainties" (their Sect. 2.5, Eq. 6). In other words, the decomposition of uncertainties allows for comparing two independent measurements with *own* (that is, independent) uncertainties, which makes conclusions regarding consistency more meaningful. The decomposition and contributing random uncertainties are thoroughly explained in Kinzel et al. (2016), their Sect. 2c.
**Changes in the manuscript:** Immler et al. (2010) has been added to Sect. 1 for a clearer motivation of our uncertainty decomposition approach (P.4, L.1f).

**14) Comment from referee:** *Pages 10-11, sections 3.4-3.5: The two sections are not directly related to any figures. Suggest to revise and combine.*
**Author's reponse:** We disagree that these two sections are not directly related to any figures/tables in the manuscript. For transparency, we believe a clear separation of all HOAPS-related uncertainties, that is systematic and random retrieval uncertainty (Sect. 3.3-3.4) and sampling uncertainty (Sect. 3.5), is appreciated. Sect. 3.3 is a main prerequisite for what is shown in Figs. 3 and 5, respectively. Sect. 4.3 (and Table 2 therein) is dedicated to only $E_{smp}$, which is first picked up

in Sect. 3.5.

**15) Comment from referee:** *Page 13, Line 10: Fig.2 is regarded as a 2-D representation of the error bar magnitude of Fig.1a. A figure showing the global pattern of HOAPS3.3 - minus - in situ needs to be provided to help interpret Fig.2.*

**Author's reponse:** The differences map points at biases, which are not linked to the random retrieval uncertainties shown in Fig 3a. Yet, the differences map (HOAPS minus in situ) has been added to the revised manuscript, where it is also commented on (P.9, L.10f,L.24f). This is already picked up in a different context (see comment #10 on this). As noted in the manuscript, the quoted passage is meant to qualitatively link the error bars in Fig. 2a to the four-dimensional (Fig. 3a) *random retrieval uncertainty* representation. Differences in their magnitudes were to be expected, as the bars in Fig. 2a include both $E_C$ and $E_{ins}$, which have been corrected for in Fig. 3a. However, the $q_a$-dependent distribution of error bar magnitudes (Fig. 2a) are very closely related to the $E_{retr,ran}$ pattern (Fig. 3a) . That is, random retrieval uncertainties are largest for subtropical ranges of $q_a$ (11-17 g kg$^{-1}$, Fig. 3a), which is mirrored in largest uncertainty bars in Fig. 2a. Likewise, these magnitudes reduce for tropical $q_a$ ranges of roughly 20 g kg$^{-1}$. Smallest magnitudes are generally found in high latitudes, where $q_a$ is smallest (below 7 g kg$^{-1}$, see Fig. 2a).  The intention was to show the spatial distribution of random uncertainty in HOAPS-3.3 $q_a$. As mentioned later on, this random uncertainty can be neglected if monthly to multi-annual averages are considered, while systematic components become the dominating source of uncertainty. Spatial maps of these long-term means of systematic uncertainties are provided in Fig. 4.

**Changes in the manuscript:** See comment #10.

**16) Comment from referee:** *Page 13, Fig. 2: The instantaneous random uncertainty map of $q_a$ (Fig.2a) has a pattern similar to the uncertainty map of $q_a$ produced by OAFlux (Yu et al. 2008, OAFlux technical report), though HOAPS3.3 has a much larger magnitude.*

**Author's reponse:** Thank you for bringing up this comparison. We agree that the error distribution shown in Yu et al. (2008) resembles our instantaneous random uncertainty distribution. Regarding uncertainty magnitudes: Yu et al. (2008) declare "mean errors" as monthly mean standard deviations (std) (time period: 1958-2006). This definition considerably differs from our approach. Furthermore, it remains unclear as to how this std is derived. Apparently, several data sets contribute to its estimation (NCEP1, NCEP2, ERA40, satellites), which may be the cause for lower magnitudes shown in their Fig. 21. Whereas our uncertainty estimates are exclusively HOAPS-related (that is, related to only one data record), the error estimation presented in Yu et al. (2008) does not clarify as to how the global error distribution includes contributions by the individual data sets.

**17) Comment from referee:** *Page 14, Line 3: In addition to Table 2, please add a zonal-mean average of the monthly mean sampling uncertainties to show the latitudinal distribution of the uncertainties.*

**Author's reponse:** We investigated the latitudinal dependency of all sampling uncertainties. Due to the large averaging time period (monthly means), there is hardly any zonal dependency evident in any of the parameters (not shown). This was to be expected, as a differentiation between tropical and extratropical buoys for quantifying monthly mean sampling uncertainies did not reveal differences in uncertainty magnitudes (see end of Sect. 3.5). As indicated in Table 2, sampling uncertainties averaged over such long time scales only show a dependency on the amount of orbiting platforms. However, this effect is not seen in the zonal means, as at least three instruments were in operational mode between 1995-2008.

**Changes in the manuscript:** A comment has been included into the revised manuscript (P.12, L.26f) that no latitudinal dependency of the sampling uncertainties exists on the monthly mean basis.

**18) Comment from referee:** *Page 14, Line 13: How is $E_{clim}$ defined? Please provide a mathematical expression of $E_{clim}$.*
**Author's reponse:** Please refer to comment #7 on this.
**Changes in the manuscript:** Please refer to comment #7 on this.

**19) Comment from referee:** *Page 14, Line 15: "Figures 3a-e can also be treated as the systematic uncertainty distribution". What is the relation between Figures 3a-e and the mean difference map of HOAPS-3.3 minus in situ? See comment Page 13, Line 10. The maps shown in Figures 3a-e are not bias patterns, as bias has both positive and negative signs. What is the meaning of the systematic uncertainty?*
**Author's reponse:** We apologize that the current formulation may be confusing. Regarding the phrase you quoted: When averaging over 25 years (1988-2012), random and sampling uncertainties become virtually zero. This implies, given our definition of $E_{clim}$ (see comment #7 on this), that $E_{clim}$ is pratically equal to the systematic uncertainty ($E_{sys}$), which in turn is the absolute representation of the bias (see Sect. 3.2). Throughout our manuscript, we do not speak of „bias patterns", as we are characterizing *uncertainties,* which are per definition non-negative. The average of an array of biases with respect to a reference can be zero, while none of the individual match ups are actually equal. This automatically points at a non-zero uncertainty. In this regard, we agree that Fig. 4 (of revised manuscript) does not show bias patterns (unlike Fig. 1 (right) in the revised manuscript), but rather patterns of $E_{sys}$. $E_{sys}$ is therefore the *upper boundary* of the (more simple) bias distribution (see Sect. 3.2).
**Changes in the manuscript:** The $q_a$ difference map (HOAPS minus in situ) has been included into the revised manuscript (Fig. 1 (right), see comment #10 and #15 on this). It is briefly described and related to Figs. 2a (P.9, L.25). Also, the composition of Sect. 3.2 has been changed.

**20) Comment from referee:** *Page 18, Line 5: "On average, it increases by roughly 4.5 W m$^{-2}$ (4.7%) per decade…". Which term gives rise to this large increase, $q_a - q_s$ or U? The continuing increase in LHF during the "hiatus" period in the 2000s does not seem realistic from the perspective of the global water budget balance (see Robertson et al. 2014, J.Clim).*
**Author's reponse:** Thank you for bringing this up. As mentioned in Sect. 4.7, this linear LHF increase over time is picked up by numerous studies and is resolved in several climatologies. Yu et al. (2007), for example, point at an OA Flux LHF increase of 9 W m$^{-2}$ over a time period of 22 years (1981-2002), which closely resembles our linear trend estimate. Our trend analysis includes a strong negative offset in HOAPS LHF during 1991. As pointed out in the manuscript (P.18, L.25ff), this is associated with retrieval issues related to the Mount Pinatubo eruption and is therefore an artificial signal. If this is solved, as has been done for the latest HOAPS version, HOAPS 4.0 (Andersson et al., 2017), the offset is smaller, which ultimately reduces the linear trend. Also, possibly related to the hiatus, global mean HOAPS LHF slightly decrease after 2008. GSSTF3 also exhibits an LHF increase up to 2007/8 (which is even stronger than that of HOAPS) and a subsequent decrease (see Robertson et al. (2014), their Fig.2b and Fig.8). Regarding the increase up to 2008, the same conclusion may be drawn for SeaFlux (Robertson et al. (2014), their Fig. 2c). As to the cause of the LHF increase: Q-term analysis indicates that linear trends of both U and $q_s$ are positive, whereas that of $q_a$ is negative. In consequence, both U and ($q_s-q_a$) give rise to the observed LHF increase. For the time period of 1988-2005, this also becomes evident in Iwasaki et al. (2014), their Fig. 9.

**21) Comment from referee:** *Page 19, Line 14: Remove the sentence. Aren't the uncertainty estimates supposed to be a common practice for all gridded products?*
**Author's reponse:** We think that it is appropriate to include this sentence in our manuscript, as we are not aware of any other satellite climate data set with such an (extensive) uncertainty characterization. We certainly agree that this should be a common practice in the future. It seems, however, that HOAPS-3.3 (and HOAPS 4.0, by now) leads the way.

**Cited studies:**

**Andersson,** A. Graw, K., Schröder, M., Fennig, K., Liman, J., Bakan, S., Hollmann, R.and Klepp, C.: Hamburg Ocean Atmosphere Parameters and Fluxes from Satellite Data - HOAPS 4.0, Satellite Application Facility on Climate Monitoring (CM SAF), doi:10.5676/EUM\_SAF\_CM/HOAPS/V002, 2017.

**Bentamy**, A., Katsaros, K. B., Mestas-Nuñez, A. M., Drennan, W. M., Forde, E. B., and Roquet, H.: Satellite Estimates of Wind Speed and Latent Heat Flux over the Global Oceans, J. Climate, 16, 637–656, doi:10.1175/1520-0442(2003)016<0637:SEOWSA>2.0.CO;2, 2003.

**Brodeau,** L., Barnier, B., Gulev, S. K., and Woods, C.: Climatologically Significant Effects of Some Approximations in the Bulk Paramererizations of Turbulent Air-Sea Fluxes, Journal of Physical Oceanography, 47, 5–28, doi:10.1175/JPO-D-16-0169.1, 2017.

**Immler**, F. J., Dykema, J., Gardiner, T., Whiteman, D. N., Thorne, P. W., and Vömel, H.: Reference Quality Upper-Air Measurements: guidance for developing GRUAN data products, Atmospheric Measurement Techniques, 3, 1217-1231, doi: 10.5194/amt-3-1217-2010, 2010

**International Vocabulary of Metrology** – Basic and General Concepts and Associated Terms (VIM 3rd edition), JCGM, 200, 2012

**Iwasaki, S.,** Kubota, M., and Watabe, T.: Assessment of various global freshwater flux products for the global ice-free oceans, Remote Sensing of Environment, 140, 549-561, doi: 10.1016/j.rse.2013.09.026, 2014.

**Robertson**, F. R., Bosilovich, M. G., Roberts, J. B., Reichle, R. H., Adler, R., Ricciardulli, L., Berg, W., and Huffman, G. J: Consistency of Estimates Global Water Cycle Variations over the Satellite Era, Journal of Climate, 27, 6135-6154, doi: 10.1175/JCLI-D-13-00384.1, 2014

**Schlüssel**, P.: Satellite Remote Sensing of Evaporation over Sea, in: Radiation andWater in the Climate System: Remote measurements, Vol. 45, NATO ASI Series, pp. 431–461, Springer-Verlag, Berlin, Germany, 1996.

**Wells, N.** And King-Hele, S.: Parameterization of tropical ocean heat flux, Quart. J. Roy. Meteor. Soc., 116, 1213–1224, doi: 10.1002/qj.49711649511, 1990

**Yu,** L., Xiangze, J., and Weller, R.A.: Multideace Global Flux Datasets from the Objectively Analyzed Air-sea Fluxes (OAFlux) Project: Latent and Sensible Heat Fluxes, Ocean Evaporeation, and Related Surface Meteorological Variables, OAFlux Project Technical Report, OA-2008-01, WHOI, 2008.

---

## Author Comment (AC3) · 1 Dec 2017

**Author's response to the general comments from referee III:**

Thank you for revealing your valuable criticism regarding the manuscript. Below, please find our responses to your specific comments, along with the implemented changes to our manuscript. All page and line numbers as well as figure numbering refer to the *revised* manuscript. Note specifically that the figure numbering has changed during the review process.

**MAIN COMMENTS:**

**1 ) Main comment from referee:**

a) The paper is hard to read. Often it requires re-reading a paragraph a number of times, to understand. It also has to do with the structure of the paper. It would help to define the main methodology of the data analysis and to have this as the main thread throughout the paper.

I think I understand the methodology, but I am still not sure.
Let me explain my interpretation of the method:
(i) Four dimensional look-up tables (LUT) are created of co-located data, so differences between data sets are stratified according to $q_a$, U, SST, and wvpa.
(ii) The mean difference between HOAPS and insitu data are interpreted as biases.
(iii) The variances of the difference are used for the triple co-location method, resulting in error estimates.
(iv) This results in LUT's of biases and random error estimates.
(v) In applications (e.g. global maps of mean and random error of $q_a$) the observations of $q_a$, U, SST and wvpa point to the table and provide errors of each observation. These can be averaged to obtain the desired map.

b) I feel that it would be helpful to describe upfront that this is the general methodology and follow it throughout the paper. So this would lead to 3 main sections in the paper: (i) Description of the methodology, (ii) Results of the methodology, i.e statistics on the LUT data, and (iii) Application to HOAPS evaporation. In case I am completely wrong on the interpretation of the paper, there is even more reason to be clear about the methodology.

c) Another question is: what is the main result of the paper? If my interpretation is correct, then the 4-dimensional table of error estimates is the main result, because it would allow a user to make estimates of anything he/she is interested in (e.g. monthly averages, daily averages, or El-Nino years). So it is worth thinking about communicating this 4D table to the users. Most of the current paper is about applying the methodology, but these are in fact just examples.

**Author's reponse:**

Related to a) We have restructured the paper to be more clear about the methodology. The introduction has been rearranged and shortened (see also specific comment #1 below) for a much clearer understanding of the motivation, the benefit, and the structure of the paper. We have furthermore appended a flowchart to this document, which guides you through the individual steps of data processing, intermediate data products, and resulting HOAPS uncertainty measures.

Regarding your methodology interpretation (the flowchart assists):
(i) Correct.
(ii) Yes, differences of paired collocations are considered as biases. Depending on U, $q_a$, SST, and wvpa, these single biases are assigned to one of the $20^4$ bins. Once all collocations have been

assigned to a bin, bin-averaged systematic uncertainties are computed based on the *absolute* differences of all assigned biases. That is, we consider the upper end of bias considerations (see comment #10 on this).

(iii) Exactly. Although the variances of differences, applied to the triplets, only help to decompose the *random* uncertainty estimates to end up with HOAPS-related (that is, retrieva-related) random uncertainty estimates.

(iv) The LUTs of systematic and (uncorrected) random uncertainties already result from i). iii) helps to decompose the random uncertainty components, isolate the retrieval-related part, and finanly correct the random uncertainty LUTs.

(v) Yes, this is correct.

Related to b) Regarding your proposed three main sections: We believe that this has already been done in a similar manner and is reflected in the numbering of the Section: Methodology (Sect. 3) and Results/Applications (Sect. 4). We do not want to dedicate „Results of the methodology" an own section, as we think that it belongs to the methodological part of the manuscript (Sects. 3.3, 3.4).

However, we agree that the submitted manuscript was not structured clearly enough. This shortcoming has been improved (see comment related to a) above).

Related to c) We agree that *one* of the main outcomes of the manuscript is the benefit of the multi-dimensional bias analyses. Particularly because the approach can be easily transferred to other satellite retrievals and potentially also other remotely sensed parameters. The approach itself should be stressed more clearly in the conclusion. Communicating our specific LUTs to the users is not helpful, though, as they are tailored to HOAPS-3.3 (due to the double collocations described in Sect. 3.1). The results of applying an updated version of the LUTs to instantaneous HOAPS data are implemented in the most recent HOAPS 4.0 Version (Andersson et al. (2017)) in form of systematic and random uncertainties.

We believe that the application of the uncertainty characterization approach is equally important, as it leads to uncertainty estimates for a widely used data record. On the one hand, none of the remaining LHF-related satellite climatologies are equipped with such estimates. On the other hand, Sect. 4 demonstrates a variety of different approaches for illustrating the uncertainties and allows for identifying regions where uncertainties in the satellite retrieval are an issue and need to be accounted for.

The focus of this paper is therefore twofold: 1) describing the method and 2) applying the method to arrive at HOAPS-3.3 uncertainty estimates.

**Changes in the manuscript:**

Related to a and b) The whole manuscript has been revised for a clearer reading experience. This specifically targets Sects. 1 and 3. The last paragraph of Sect. 1 now guides the reader through the manuscript step by step. Section 3.3. and 3.4 have been swapped to be consistent with the sequence of analyses.

Related to c) The benefit of the multi-dimensional bias analyses for uncertainty characterizations has been highlighted more clearly in Sect. 5 (P.20, L.21ff).
* * *
**2 ) Main comment from referee:**

Estimation of biases is non-trivial. In fact this is very important because, as the authors point out, for long term averages the systematic errors dominate.

My concern is two-fold:
a) I have the feeling that it is assumed that DWD-ICOADS data is bias-free? If this is the basis for the bias estimates, then it deserves more discussion also in view of what has been published in literature.
b) Fig. 1 is used as an example to illustrate the estimation of biases. However, it is likely that artificial biases occur in binned scatter plots of noisy data if correlated variables are used on abscissa and ordinate. This applies to Fig. 1a where hair(HOAPS) is used on both vertical and horizontal axes. It also applies to hair versus wind because these variables are correlated due to the physics of the mixing (more wind brings hair closer to the surface value). To check, one could e.g. bin the differences of Fig. 1a in classes of hair(insitu). Also hair(insitu) is noisy because it has large representativeness errors (point observation, whereas HOAPS has a large footprint).

c) Finally, if one can be confident about the bias estimation, then it should also be trivial to apply a bias correction to HOAPS. This would just leave the uncertainty in C_E which is a parametrization constant used for satellite as well as in-situ data. Please discuss.

**Author's reponse:**

Related to a) It is correct that we assume the DWD-ICOADS data base to be bias free (see last paragraph in Sect. 2.2 of submitted manuscript). Our filtering procedure ensures that only high-quality in situ data is used for collocation analysis. Systematic effects of known origin are thought to have been removed or at least minimized within the quality checking procedure at the Marine Climate Data Center of DWD. Other systematic uncertainties like differing sensor heights and cool skin effects have been eliminated prior to our analysis due to sensor height corrections using in situ platform meta data (U) and cool skin corrections ($q_s$). We are aware of the fact that no ground "truth" exists, but are confident that our extensive data base is the best ground "reference" available. Freeman et al. (2016) present a great overview of the variety of ICOADS applications, which also include the calibration and validation of satellite data (e.g. Bentamy et al. (2003), Bentamy et al. (2013), Jackson et al. (2009), Jackson and Wick (2010)).
It should be kept in mind that our systematic uncertainty estimates represent the upper limit of a more simple bias estimation. Assuming a bias free ground reference therefore does not violate our conclusions, although a small contribution to the systematic uncertainties may be caused by the in situ reference.
One could argue that our uncertainty estimates in regions of poor in situ data coverage are questionable. However, as picked up in Sect. 3.2, we overcome the regional dependency by characterizing uncertainties as a function of ambient atmospheric conditions. Poor in situ data densities are therefore of secondary importance, as their ambient atmospheric conditions may be similar in regions with considerably more match ups.

Related to b) Thank you for the suggestion to investigate the one-dimensional patterns of dq as a function of the in situ source. We exemplarily performed this analysis for 2001 with approximately 1.8 million match ups. We compared the magnitudes of the mean 5-percentiles, which (in case of HOAPS) are illustrated as black squares in Fig. 2. For U, $q_s$, and $q_a$, our results indicate that in 80% of all match ups (i.e., excluding the margins), relative differences between HOAPS and in situ mean 5-percentiles range between ± 6-10%, which we consider as negligible. We presume that a two- instead of one-sided regression approach would lead to even more robust 5-percentile means. Towards the margins of the distributions, relative differences become larger. We believe that this does not have a noteworthy impact on the four-dimensional analyses, as the biases in one-dimensional space may become smaller or even cancel out when the remaining three atmospheric state parameters are considered concurrently.
Independent of this, biases as a function of in situ LHF-related parameters cannot be investigated in four-dimensional space, as vertically integrated water vapour ("wvpa"), an important indicator for

the prevalent atmospheric condition, is not available from in situ measurements. This would lead to an undesirable simplification of our uncertainty analysis approach. Additionally, our match up data base only lasts until 2008. In consequence, no uncertainties could be assigned to pixel level HOAPS data from 2008 onwards, if the multi-dimensional bias approach was based on in situ data.

Related to c) Regarding a bias correction of HOAPS data: Our approach aims at characterizing uncertainties inherent to HOAPS. This allows users to implement this information into their analyses and arrive at appropriate conclusions. We have further emphasized the benefit of such estimates in the revised version of Sect. 1. The focus is therefore not put on bias correction with respect to DWD-ICOADS. A sustainable consequence of large uncertainties should in fact point at the need of modifying the retrieval algorithm instead of bias correcting the data. It is our impression that a bias correction is feasible, if a constant bias (in terms of dependent variables, region, or time) is present relative to a fiducial reference. Such a reference is not available at present.

**Changes in the manuscript:**

regarding a) Freeman et al. (2016) is picked up in context of describing the in situ data base (P.6, L.18f). Furthermore, some further references are given regarding our assumption of bias-free ICOADS measurements (P.8, L.3ff).

regarding b) We briefly mention the artificial biases due to correlating variables and conclude that two-sided regression analyses could reduce these spurious biases (P.9, L.17ff).
* * *
**SPECIFIC COMMENTS:**

**1 ) Comment from referee:**
Section 1: Although well written, the introduction is rather long and contains sometimes fluffy language. No reference is made to an earlier study by Kinzel et al. (2016). What is new compared to earlier work? Reference is made to other data sets and to studies that provide error estimates. However, nothing is said about published error estimation methods.
**Author's reponse:** We agree that the introduction of the submitted manuscript is too long. We have restructured Sect. 1 following your suggestions and believe that this essentially improved the manuscript (see also "main comment 1" on this). Furthermore, Kinzel et al. (2016) has been included in the revised manuscript to point at the  random uncertainty decomposition approach. We agree it is important to distinguish between earlier work and new aspects of this manuscript. This also includes a statement regarding earlier error estimation methods and those present in our manuscript.
**Changes in the manuscript:** The first 24 lines have been considerably shortened.
The whole introduction has been restructured to be clearer about the motivation and benefit of our study. It now clearly differentiates between earlier approaches (that is, mostly intercomparison studies, P.3, L.25ff) and the novelty of our uncertainty characterization (that is, uncertainty estimates that are exclusively related to a specific data set, in particular HOAPS, P.4, L.1f.; P.4, L.14ff). At the same time, we highlight the new aspects of our approach (e.g. four-dimensional LUTs, P.4, L.20f). In this regard, Kinzel et al. (2016) has been included into the revised manuscript and is put into context (P.4, L.10f, L.21f). The advantage of multi-dimensional LUTs has been included into the abstract (P.1, L.7f).

**2 ) Comment from referee:** Page 5, Line 32: The sentence with "The latter depends" suggests that it refers to $q_a$ in the sentence before, but what it intends to say is that the COARE algorithm needs stability and that specific assumptions are made. Please rephrase.
**Author's reponse:** The wording was chosen on purpose, as we wanted to point out that the

saturation vapour pressure and hence $q_a$ depends on the surface air temperature. However, we agree that this may be confusing and the focus should be put on the stability calculation.

**Changes in the manuscript:** We changed the wording to "It includes atmospheric stability calculations, which necessitate surface air temperatures as input. These are estimated by assuming..." (P.6, L.7f).

**3 ) Comment from referee:** Page 6, Line 20-24: The non-correction of $q_a$ for measuring height is confusing. Why not using the real measuring height in the bulk formula? Perhaps it is possible to say in one sentence what the results are of the height difference effects as estimated by Kent et al. (2014).

**Author's reponse:** Prytherch et al. (2014) and Kinzel et al. (2016) point at the disadvantages related to $q_a$ height corrections. We agree that a statement regarding the height correction effect is useful. Kent et al. (2014) quantify the height correction effect to be 0.11 g kg$^{-1}$ for the time period 1971-2006, owing to the continuously increasing measurement platform heights. However, this effect is masked by bias corrections associated with measurement techniques, which are thought to be 2-3 times larger.

**Changes in the manuscript:** Results by Kent et al. (2014) regarding the height difference effects are briefly mentioned (P.7, L.3ff).

**4 ) Comment from referee:** Page 7, Line 13: Cool skin corrections are applied to in situ observation but not to HOAPS-3.3 SST (AVHRR based). This makes sense in priciple because AVHRR measures the skin temperature. However, there must be a calibration procedure of AVHRR, which is probably against bulk SST data. So, what does calibrated AVHRR data represent, bulk or skin SST?

**Author's reponse:** Thank you for bringing this up. Indeed, AVHRR was calibrated against bulk SST. Formally, this would necessitate a cold skin correction. However, compared to OISST (Reynolds et al. (2007)), AVHRR has a cold bias of unknown origin, which is in the order of the skin correction. We therefore refrained from performing the correction and consider the AVHRR SST as a skin SST. Note that this cold bias problem is overcome in HOAPS 4.0 (Andersson et al. (2017), which is based on OISST. For the HOAPS 4.0 retrieval (Andersson et al. (2017)), OISST is corrected for the cold skin effect.

**5 ) Comment from referee:** Pages 4-5 section 2.1: It would be informative to mention pixel size of the microwave sensors.

**Author's reponse:** Yes, we agree.

**Changes in the manuscript:** Pixel sizes have been included into Sect. 2.1 of the revised manuscript (P.5, L.11-13).

**6 ) Comment from referee:** Page 8, Line 11: The sentence "Figure 1a overestimates ...." is confusing. Formally it is correct, but, after reading the first time it suggests that the biases range from 7-12 g/kg and that the plot is for the inner tropics.

**Author's reponse:** Indeed, this may be misunderstood.

**Changes in the manuscript:** The wording has been changed in the revised manuscript to: "For $q_a$ values between 7-12g kg$^{-1}$ , HOAPS-3.3 overestimates near-surface specific humidities (see Figure 2a). Overestimations are also observed in the inner tropics, where $q_a$ is in the order of 20 g kg$^{-1}$" (P.9, L6f).

**7 ) Comment from referee:** Page 8, Line 17: The expression "over-(under-)estimated" is perhaps better than "over-(under-)represented"

**Author's reponse:** Thank you for this suggestion.

**Changes in the manuscript:** "over-(under-)represented has been replaced by "over-(under-)estimated" (P.9, L.12).

**8 ) Comment from referee:** Scatter plots in Fig.1: In all the plots except (c) the variables on the vertical axis are correlated with the variable of the horizontal axis. This is most obvious for Fig. (a) where hair-HOAPS is used in both abscissa and ordinate. In such cases the binning according to one axis can show biases that are not necessarily real. Whether this is really the case can be easily demonstrated by making the same plot but now with hair-insitu on the horizontal axis. Similarly unrealistic bias may be seen in (b) and (d) because wind and wvpa are derived with from the same satellite channels and therefore correlate with hair-HOAPS. Please discuss.

**Author's reponse:** We assume that "hair-insitu" means "$q_a$(in situ)" and not the mathematical difference between HOAPS and in situ $q_a$? We are aware of the correlation between the individual variables. The aspect of correlating variables is an important remark, which we thoroughly discuss in context of the "main comment 2" (part b), see further above). In fact, this is fundamental for our multi-dimensional approach: characterizing systematic and random uncertainty estimates of U, $q_s$, and $q_a$ as a function of atmospheric state parameters, which (as we believe) have an impact on the parameters themselves. Specifically regarding Figure 2d): wvpa is not available from in situ measurements, which is why a bias dependency on in situ wvpa cannot be investigated.

**Changes in the manuscript:** See "main comment 2" (part b) further above.

**9 ) Comment from referee:** Page 9, Line 21: Please specify what "even stronger winds" are.

**Author's reponse:** "stronger wind" mean wind speeds exceeding 20 m s$^{-1}$.

**Changes in the manuscript:** The wording has been changed in the revised manuscript (P.10, L.9).

**10 ) Comment from referee:** Page 9, Lines 24-26: This paragraph is hard to read. After reading, a number of times times, I think I understand. Is it not better to say: "Our goal is to document the upper bound of the bias and therefore we take the absolute value of the possible systematic error in CE"?

**Author's reponse:** We agree that this paragraph is somewhat confusing and out of place. It has been moved further up into the appropriate context.

**Changes in the manuscript:** The wording has been modified and has been moved further up into the appropriate context (P.10, L.17ff).

**11 ) Comment from referee:** Page 10, line 15 and page 11, line 7: I suggest to replace "Next to" by "In addition to"

**Author's reponse:** Thank you for this suggestion.

**Changes in the manuscript:** The wording has been changed in the revised manuscript (P.12, L.12).

**12 ) Comment from referee:** Page 11, section 3.5: This section is hard to read. If I understand correctly, it addresses the question: Does it matter for the averages that the satellites sample the ocean at particular times of the day only, given that a diurnal cycle may be present? The authors investigate by looking at buoy data and by comparing averages that cover the full diurnal cycle with samples at satellite overpass times only. Part of the confusion is because it mentions spatial sampling, but I don't think this section covers that? Please simplify for clarity.

**Author's reponse:** Exactly. For the monthly mean HOAPS product (HOAPS-G), sampling uncertainties need to be quantified because of the diurnal cycle of the geopyhsical parameters. Due to the sun-synchronous satellite overflights, diurnal cycles or frontal passages are likely to be missed. This will affect the monthly mean averages. We agree that the the term "spatial sampling" is misleading, as we only cover the temporal sampling issue.

**Changes in the manuscript:** The aspect of "spatial sampling uncertainties" has been removed from the revised manuscript to avoid confusion.

**13 ) Comment from referee:** page 13, Lines 9-11: I am not sure that it is helpful here to refer to Fig. 1a, because it is showing the combination of E_ins($q_a$_a) and E_retr($q_a$), which is different from

Fig. 2a. The authors point this out but instead of clarifying something it confuses.
**Author's reponse:** We agree that this may be confusing.
**Changes in the manuscript:** This section has been shortened to become more clear. (P. 14, L.18ff)

**14 ) Comment from referee:** Page 13, 23: Suggestion: replace "merely" by "only"
**Author's reponse:** Thank you.
**Changes in the manuscript:** "merely" has been replaced by "only". (P.14, L.31).

**15 ) Comment from referee:** Page 13, Line 24: What is meant by "local minimum in that region for q_a"? E_retr(q_a) has a maximum over the warm pool.
**Author's reponse:** This is a mistake in our manuscript, thank you for pointing this out. We wanted to point at the $q_a$ random retrieval uncertainty, not $q_a$ itself.
**Changes in the manuscript:** the wording has been changed (P.14, L.31f).

**16 ) Comment from referee:** Page 13, Line 29: In the sentence "Respective values partly exceed 50 W/m$^2$", what is meant by "respective" and "partly"? Do the authors mean: "In these areas, values are found in excess of 50 W/m$^2$"?
**Author's reponse:** Yes, this is correct.
**Changes in the manuscript:** The wording has been changed in the revised manuscript.  (P.15, L.3).

**17 ) Comment from referee:** Page 14, Line 33: "direct eddy covariance" is not wind speed.
**Author's reponse:** Sorry for not being correct here. The wind stresses are based on inertial-dissipation methods. Together with eddy covariance based LHF, the turbulent fluxes of a variety of satellite, reanalysis, and combined products are evaluated.
**Changes in the manuscript:** The wording has been changed. (P.15, L.31ff).

**18 ) Comment from referee:** Page 16, Lines 1-2: This is an interesting example, where it is explained that $q_a$ retrievals may be in error because of dry air advection. However, it is not clear how the systematic error analysis picks up the area of the Agulhas current. The systematic error estimation is entirely driven by U, $q_a$ and SST and wvpa (if I understand correctly).
**Author's reponse:** It is correct that the systematic uncertainty estimation is entirely driven by combinations of ambient U, $q_a$, SST, and wvpa. Our multi-dimensional bias approach does not point at specific regions. This implies that we cannot be 100% certain that the observed uncertainties over the Agulhas region are exclusively associated with local retrieval issues. In general, match ups over a region contribute to the look up tables (LUTs), which implies these regions are somewhat mirrored in the LUTs. However, they are not explicitly resolved.
The following serves to explain how the LUTs pick up the Agulhas region: Figure 1 (left) indicates that numerous collocations between buoys/ships and satellite exist in this area, which is characterized by a unique combination of ambient U, $q_a$, SST, and wvpa. In case of the mentioned dry cold air outbreaks from the South, $q_a$ will be anomalously low and hence $q_s$-$q_a$ and LHF anomalously large. According to Santorelli et al. (2011), satellite retrievals seem to encounter difficulties with these dry cold air outbreaks, which implies that they will not capture $q_a$ correctly. This would for example be seen when investigating $dq_a$. That is, differences between satellite and in situ $q_a$ would be negative, which directly impacts our four-dimensional uncertainty analysis. In conclusion, repetitive retrieval issues over a specific regions will be manifested in the LUTs and will eventually be seen in systematic uncertainty maps. At the same time, underestimated $q_a$ along the Agulhas Current contribute to an increase in the random uncertainty component of the LUTs.

**19 ) Comment from referee:** Section 4.6 and Fig. 4: Here both systematic and random errors are discussed region by region and climatologically versus January/July. Earlier in the paper it was concluded that the random errors were small compared to the systematic errors. However in Fig. 4 the random errors are larger than the systematic errors. Furthermore I would expect that the

climatological data (I assume averaged over the entire period) has much more data than the January or July data and therefore much smaller random errors.

**Author's reponse:** As mentioned in Sect. 4.6, the error bars in Fig. 5 point at *instantaneous* random uncertainties (such as those shown in Fig. 3). The idea is to show the maximum uncertainty to be expected for a specific region and season on an *instantaneous* basis. This approach allows for illustrating random uncertainties, as they often even exceed the systematic counterpart for pixel-level data, as is seen when comparing Fig. 3 to Fig. 4. Fig. 3 shows averaged *instantaneous* random uncertainties as a function of region and time. If properly scaled according to the considered period of time, they decease with increasing time period and become insignificant at monthly or multi-annual (that is, climatological) time scale. Keeping this in mind, this also answers the question as to the smaller random uncertainties for multi-annual mean (1988-2012) compared to seasonal means (1988-2012): The difference in error bar magnitudes is not related to averaging periods, as these are averaged *instantaneous* random uncertainties as a function of region and time. We agree, however, that this is not clearly stated in the manuscript.

**Changes in the manuscript:** The wording as been modified in the revised manuscript to clarify what is shown in Fig. 5. (P.18, L.29-33) This also targets the caption of Fig. 5.

**20 ) Comment from referee:** Page 18, Line 31: Please replace "outperforms" by "exceeds"
**Author's reponse:** Thank you for this suggestion.
**Changes in the manuscript:** This section has been considerably shortened. The phrase is no longer included in the revised manuscript.
* * *
**Cited:**

**Andersson,** A. Graw, K., Schröder, M., Fennig, K., Liman, J., Bakan, S., Hollmann, R.and Klepp, C.: Hamburg Ocean Atmosphere Parameters and Fluxes from Satellite Data - HOAPS 4.0, Satellite Application Facility on Climate Monitoring (CM SAF), doi:10.5676/EUM\_SAF\_CM/HOAPS/V002, 2017.

**Freeman**, E., Woodruff, S. D., Worley, S. J., Lubker, S. J., Kent, E. C., Angel, W. E., Berry, D. I., Brohan, P., Eastman, R., Gates, L., Gloeden, W., Ji, Z., Lawrimore, J., Rayner, N. A., Rosenhagen, G., and Smith, S. R.: ICOADS Release 3.0: a major update to the historical marine climate record, Int. J. Climatol., p. in press, doi:10.1002/joc.4775, 2016.

**Kinzel,** J., Fennig, K., Schröder, M., Andersson, A., Bumke, K., and Hollmann, R.: Decomposition of Random Errors Inherent to HOAPS-3.2 Near-Surface Humidity Estimates Using Multiple Triple Collocation Analysis, J. Atmos. Oceanic Technol., 33, 1455–1471, doi:10.1175/JTECH-D-15-0122.1, 2016.

**Reynolds**, R. W., Smith, T. M., Liu, C., Chelton, D. B., Casey, K., and Schlax, M. G.: Daily High-Resolution-Blended Analyses for Sea Surface Temperature, J. Climate, 20, 5473–5496, doi:10.1175/2007JCLI1824.1, 2007.

---

## Author Comment (AC6) · 1 Dec 2017

Please find attached a PDF document showing the revised manuscript. Towards the end, you will also find a document showing the changes that have been implemented into the latest version. Both documents have been merged.

Please also note the supplement to this comment:
https://www.atmos-meas-tech-discuss.net/amt-2017-176/amt-2017-176-AC6-supplement.pdf
* * *

---

## Referee Report (RR1)

The revised version of the paper entitled "Uncertainty Characterization of HOAPS-3.3 Latent Heat Flux Related Parameters" by Julian Liman, Marc Schroder, Karsten Fenning, Axel Andersson, and Raimer Hollman and referenced as amt-2017-176, improves significantly the original version. The authors provides detailed responses to all my comments and suggestions. I think the paper can be accepted for AMT publication after minor revision.

Specific comments:

1. Abstract, L.1   Why do you use not "flux" but "fluxes"?   Is it consistent with " one of "?

2. P.3,L.26   Kubota et al.(2003) carried out intercomparison studies before Chou et al.(2004).

3. P.15,L.13

   Eclim is defined grid point wise as the mean root mean squared sum of instantaneous   $E_{sys}$,   $E^{ran}_{ret}r$, and   $E_{smp}$ between 1988-2012.

   P. 18,L. 32

      they are independent of each other

   Are two sentences consistent?

4. P.18, L.22

      20Wm-2   (33Wm-2 ), respectively. These estimates are 20%(11 %)

      ----$\rightarrow$ 20Wm-2 and 33Wm-2 , respectively. These estimates are 20% and 11 %

---

## Author Response (AR2)

**Author's response to the general comments from referee I:**

Thank you for your comments regarding our revised manuscript. Below, please find the changes that have been implemented into its latest version.

**1) Comment from referee:** Abstract, L.1 Why do you use not "flux" but "fluxes"? Is it consistent with "one of"?
**Author's response:** Thank you for this suggestion.
**Changes in the manuscript:** „fluxes" has been changed to „flux" (P. 1, L. 1)

**2) Comment from referee:** P.3,L.26 Kubota et al.(2003) carried out intercomparison studies before Chou et al.(2004).
**Author's response:** This is correct.
**Changes in the manuscript:** Kubota et al. (2003) has been included in the revised manuscript (P. 3, L. 26).

**3) Comment from referee:** $E_{clim}$ is defined grid point wise as the mean root mean squared sum of instantaneous $E_{sys}$, $E_{ran,rel}$, and $E_{smp}$ between 1988-2012. P. 18,L. 32 they are independent of each other. Are two sentences consistent?
**Author's response:**  Thank you for bringing this up. Both sentences are consistent. There is no physical explanation for why there should be a dependency among the three uncertainty components. The International Vocabulary of Metrology (VIM) gives a great overview on the definitions of systematic and random uncertainties by clearly emphasizing their differences. If they were not independent, a separation (as we performed) would not have been possible. Sampling uncertainties occur, if a parameter having a clear diurnal cycle is repeatedly sampled at fixed local times. Even if they were zero, this would not impact both systematic and random uncertainties, as they have a completely different origin. Formally, independent uncertainty components add up in the shape of RMSEs; this explains our definition of $E_{clim}$ being the RSME of $E_{sys}$, $E_{ran,rel}$, and $E_{smp}$.

**4) Comment from referee:** P.18, L.22 $20Wm^{-2}$ ($33Wm^{-2}$), respectively. These estimates are 20%(11 %) --> $20Wm^{-2}$ and $33Wm^{-2}$, respectively. These estimates are 20% and 11 %.
**Author's response:** We agree that your suggestion improves the structure of the sentence.
**Changes in the manuscript:** We rephrased the sentence according to the referee's suggestion (P. 18, L. 34).

**Cited study:**
**International Vocabulary of Metrology** – Basic and General Concepts and Associated Terms (VIM 3rd edition), JCGM, 200, 2012

**Author's response to the general comments from referee III:**

Thank you again for submitting your valuable comments regarding the present manuscript. Below, please find our responses to your specific comments, along with the implemented changes to our manuscript.

**1) Comment from referee:** Although I can live with this manuscript as is, to improve accessibility of the paper, I suggest to rewrite the methodology part of the abstract (say page 1, lines 4 to 10), the last paragraph of the introduction (page 4, lines 19 to 30), and the introduction of the methodology section (page 8, lines 12 to 20) from scratch.
**Author's response:** We agree that some of the terminology is not intuitive enough for readers. We therefore rewrote the suggested passages by explicitly explaining the used jargon in a clearer way. At the same time, the revised paragraphs do not contain too much details, as they should be understood as an overview guidline for the subsequent subchapters.
**Changes in the manuscript:** The following paragraphs have been rewritten: P. 1, L. 4-14; P. 4, L. 19-33; P. 5, L. 3-7; P. 8, L. 20-33.

**2) Comment from referee:** The reference to Kinzel et al. (2016) on page 4, line 21 is still rather obscure as part of the introduction. In the introduction I suggest to explain: what is covered in the Kinzel et al. paper and what is new in the current work.
**Author's response:** We agree that the citation seems out of place. Following comment 1), it is now embedded within the more thorough explanation of the method and related definitions. Towards the end of Sect. 1, it is again picked up to indicate in what way the current manuscript includes new aspects.
**Changes in the manuscript:** The reference of Kinzel et al. (2016) has been moved to two adequate places (P. 4, L. 2/; P. 5, L. 5).

**3) Comment from referee:** Page 9 lines 17 to 21 addresses the issue of correlated variables on absis and ordinate, which is good. I guess that on line 18, q_a, U and SST are from in situ observations rather than satellite, but it is not mentioned explicitly. I am confused, because the same symbols are used as on line 2.
**Author's response:** This is absolutely correct, thank you for pointing this out.
**Changes in the manuscript:** A note has been added to the revised manuscript (P. 9, L. 30), indicating that the abscissa shows the *in situ* observations.

[revised manuscript text omitted]

---

## Author Response (AR3)

**Author's response to the comments from Editor:**

Dear Ad Stoffelen,

thank you for your comments regarding our revised manuscript. Below, please find our responses and changes that have been implemented into its latest version.

**1) Comment from referee:** The methodology appears to rely on conditional sampling, e.g., if x and y measure truth t, then the mean of y-x for given x is evaluated and interpreted as the bias (or SD) of y versus t. This is incorrect of x has an error. The error of x will broaden the distribution of x w.r.t. t and therefore extreme x are larger than extreme t, such that y should be biased against x, even though x and y have only random error w.r.t. t (See, e.g., Stoffelen, 1998); See, inter alia, Figure 2.
**Author's response:** Thank you for pointing this out. The occurrence of these artifical biases was already picked up by Referee III during the first iteration of the manuscript (P. 9, L.29-32). To assess their level of influence on the uncertainty analysis, the biases (for $q_a$ shown in Fig. 2, but also for U and $q_s$) were illustrated as a function of the *in situ* source, exemplarily for 2001. Results indicate that in 80% of all match ups, relative difference between HOAPS and the in situ mean 5-percentiles (indicated by black squares, respectively) range between ± 6-10%, which we consider as negligible. Differences in the tail regimes (below/above the $10^{th}/90^{th}$ percentile) exceed 10%. A minimization of these pseudo biases can be achieved by considering the average of both approaches (that is, HOAPS and *in situ* as abscissa variables), which is envisaged for future HOAPS uncertainty characterizations. Moreover, defining bins on the basis of percentiles (as has been done, rather than equidistant bins) contributes to constraining the pseudo biases.
Despite the occurrence of pseudo biases, we would like to add:
i) the influence of the pseudo biases, specifically in the tail regimes, becomes smaller when investigating bin-wise biases in four-dimensional space, which is fundamental to our uncertainty characterization. Speaking graphically, pair-wise biases may fall into a *neighboring* or *nearby* bin (of $20^4 = 160000$ bins in total) within the four-dimensional look up tables (LUTs). Not only do neighboring biases highly correlate; this "displacement" can even average out, once millions of match ups have been assigned to unique bins within the LUTs.
ii) as stated in Sect. 2 (P.8, L.13-18), our uncertainty estimates should be interpreted as *upper-boundary* estimates. In the tail regions, pseudo biases therefore provoke artificial increases in the HOAPS uncertainty estimates. We believe that upper-limit uncertainty estimates are more confidential for the user community compared to lower-limit estimates concealing retrieval issues.
**Changes in the manuscript:** We now point out that an investigation of biases w.r.t. the *in situ* sources is envisaged for future HOAPS versions (P.10 , L.1-2). Additionally, we cite Stoffelen (1998) when mentioning that errors in buoys can lead to pseudo biases and therefore to an increase of the HOAPS uncertainty estimates (P.9 , L.32-33).

**2) Comment from referee:** It is assumed that errors in q, U and T are independent, which appears rather odd? In areas with high SST variability, high wind variability will occur, as well as variability in humidity and air temperature. Also, in moist convection, all atmospheric parameters tend to vary simultaneously and in correlated ways. Besides correlations expected from physical processes, correlations are also expected in the simultaneous retrievals. A retrieval is an algorithm where radiance measurements are compromised in order to obtain a geophysical retrieval. Ergo, an error in U in the retrieval is likely associated with compensating errors in the other retrieved geophysical variables. Furthermore, if the same QC, cal/val and retrieval algorithms are used for two different instruments, error correlation of those multi-variable retrievals is likely too. This should be made clear.
**Author's response:** It is not clear to us, to which part of the manuscript the comments regarding the independency of errors belong to. We believe they point at the random uncertainty decomposition using triple collocation analysis. In this case, the reader is referred to Kinzel et al. (2016). Their Sect. 2c (P. 1460) throughly describes why we assume the individual uncertainty components contributing to the error models to be independent of the satellite platform. Moreover, an error correlation term explicitly contributes to the variances of differences (their Eq. 1), which is non-negligible when the individual

error terms are not independent. This error correlation term is not neglected in Kinzel et al. (2016), but is explicitly accounted for. We certainly agree that an error in e.g. U in the retrieval is likely associated with compensating errors in other geophysical retrievals. For example, parts of the random uncertainties shown in Fig. 2a (error bars) receive a systematic component in Fig. 2b (squares) (P.10, L.8-10). In fact, this motivated us to characterize the LHF-related uncertainties using a *multi-dimensional* approach, where we explicitly separate systematic from random uncertainties.

**Changes in the manuscript:** Kinzel et al. (2016) is cited in context of the error models (P.11 , L.28). Additionally, we mention that error correlation terms are explicitly accounted for when decomposing the random uncertainties (P.11, L.30).

**3) Comment from referee:** The error model is not clear formally. What is assumed to be truth and what instrumental and geophysical variations are exactly captured by the error variables?

**Author's response:** We assume that the comment targets the random uncertainty decomposition approach summarized in Sect. 3.3. As noted in the manuscript (P.11, L.18-19), technical details regarding the uncertainty decomposition are provided in Kinzel et al. (2016). Specifically regarding the error models, their Eqs. 2a-2b provide more insights. In case of *in situ* data, the only random uncertainty source is related to instrument noise ($E_{ins}$). Regarding the satellite, both random model uncertainties ($E_M$) and sensor noise ($E_N$) contribute to the random uncertainty component. When collocating, random collocation uncertainties ($E_C$) come into play. The first three listed uncertainty sources are related to the instruments, while $E_C$ is a function of the geophysical parameter and the ambient conditions. The variances of differences are then derived bin-wise, once a bias correction w.r.t. SWA-ICOADS has been performed. We believe it is reasonable to consider the *in situ* data as our bias free ground reference (P.8, L.13-18), once sensor height corrections (in case of U) and cool skin effects (in case SST) have been carried out. Again, this assumption implicates that our LHF-related uncertainty estimates should be treated as upper-limit estimates, which is already picked up towards the end of Sect. 2 (P. 8, L.15-16).

**Changes in the manuscript:** We cite Kinzel et al. (2016), specifically when mentioning the error model (P.11, L.28) (see comment 2) above).

**4) Comment from referee:** The manuscript does not appear to clearly separate statistical and geophysical effects. Of course, statistical results may be linked to physical processes, but also to coincidental occurrence of high humidity and low winds, for example of conditional binning artifacts.

**Author's response:** We are unsure as to what section the comment refers to in the manuscript and therefore provide a general answer. We agree that no clear separation as to the type of effects has been done. Our multi-dimensional bias analyses result in bin-dependent biases (systematic uncertainties) and their spread (random uncertainties). The assignment of a single bias (e.g. $q_a$ (HOAPS) minus $q_a$ (*in situ*)) to one of the $20^4$ bins depends on physical parameter dependencies. Therefore, the spanning of the four axes is geophysically motivated. The actual binning however, which results in bin-dependent biases and their spreads, is a statistical approach, as we do not explicitly solve for the parameter functionalities. We would like to point out that the scope of this manuscript lies on a general uncertainty characterization of HOAPS LHF-related parameters, irrespective of which effects cause them.

**Changes in the manuscript:** A note has been added when introducing the multi-dimensional bias approach that our motivation is geophysical, whereas the actual implementation is statistical (P.11, L.2).

**Cited studies:**

**Kinzel, J.,** Fennig, K., Schröder, M., Andersson, A., Bumke, K., and Hollmann, R.: Decomposition of Random Errors Inherent to HOAPS-3.2 Near-Surface Humidity Estimates Using Multiple Triple Collocation Analysis, J. Atmos. Oceanic Technol., 33, 1455–1471, doi: 10.1175/JTECH-D-15-0122.1, 2016.

**Stoffelen, A.**: Toward the true near-surface wind speed: Error modeling and calibration using triple collocation. J. Geophys. Res., 103 (C4), 7755–7766, doi: https://doi.org/10.1029/97JC03180, 1998.